# MAS²: Self-Generative, Self-Configuring, Self-Rectifying Multi-Agent Systems

**Kun Wang**🦌† **Guibin Zhang**🦁†, **ManKit Ye**🐦†, **Xinyu Deng**🐦, **Dongxia Wang**🐦‡,
**Xiaobin Hu**🦫, **Jinyang Guo**🚀, **Yang Liu**🦌, **Yufei Guo**☁️‡
🦌NTU 🦁NUS 🐦USTC 🐦ZJU 🚀BUAA ☁️PKU † Equal Contribution ‡ Corresponding
✉Corresponding: `wang.kun@ntu.edu.sg`, `yfguo@pku.edu.cn`, `dxwang@zju.edu.cn`

## ABSTRACT

The past two years have witnessed the meteoric rise of Large Language Model (LLM)-powered multi-agent systems (MAS), which harness collective intelligence and exhibit a remarkable trajectory toward self-evolution. This paradigm has rapidly progressed from manually engineered systems that require bespoke configuration of prompts, tools, roles, and communication protocols toward frameworks capable of automated orchestration. Yet, dominant automatic multi-agent systems, whether generated by external modules or a single LLM agent, largely adhere to a rigid "*generate-once-and-deploy*" paradigm, rendering the resulting systems brittle and ill-prepared for the dynamism and uncertainty of real-world environments. To transcend this limitation, we introduce MAS², a paradigm predicated on the principle of recursive self-generation: a multi-agent system that autonomously architects bespoke multi-agent systems for diverse problems. Technically, we devise a "*generator-implementer-rectifier*" tri-agent team capable of dynamically composing and adaptively rectifying a target agent system in response to real-time task demands. Collaborative Tree Optimization is proposed to train and specialize these meta-agents. Extensive evaluation across seven benchmarks reveals that MAS² achieves performance gains of up to $19.6\%$ over state-of-the-art MAS in complex scenarios such as deep research and code generation. Moreover, MAS² exhibits superior cross-backbone generalization, effectively leveraging previously unseen LLMs to yield improvements of up to $15.1\%$. Crucially, these gains are attained without incurring excessive token costs, as MAS² consistently resides on the Pareto frontier of cost-performance trade-offs. The source codes are available at https://github.com/yeyeyeah2/MAS2.

## 1 INTRODUCTION

Inspired by the collective intelligence observed in human societies (Minsky, 1988; Li et al., 2023a), large language model (LLM)-based multi-agent systems (MAS) have evolved into intricate ecosystems composed of multiple LLMs, tool integrations, memory modules, and communication protocols (Zhuge et al., 2024; Tran et al., 2025). In contrast to single-agent paradigms that seek a monolithic, all-powerful model capable of addressing every task in isolation, multi-agent systems harness the virtues of specialization and cooperative interaction among *heterogeneous agents*, with substantial advances across a spectrum of task domains, including scientific discovery (Ghafarollahi & Buehler, 2024; Ghareeb et al., 2025), deep research (Zhang et al., 2025d; Hu et al., 2025a), complex report generation (Yi et al., 2025), and collective reasoning (Ye et al., 2025a).

The evolution of MAS has unfolded as a clear progression from *manual design* to *full automation*. **(I) manually configured**: Early frameworks such as AutoGen (Wu et al., 2023), MetaGPT (Hong et al., 2023), and AgentVerse (Chen et al., 2023) relied entirely on hand-crafted specifications, including prompts, agent roles, tool integrations, and communication topologies. **(II) partially automated**: Subsequent efforts introduced partial automation: for example, GPTSwarm (Zhuge et al., 2024) and G-Designer (Zhang et al., 2024b) automated the design of inter-agent communication topologies, while MasRouter (Yue et al., 2025) and LLM-Selector (Chen et al., 2025a) focused on automating LLM routing. **(III) fully automatic**: Recently, the community has converged on fully automated MAS whose configurations are synthesized end-to-end and adapted online to domain and instance

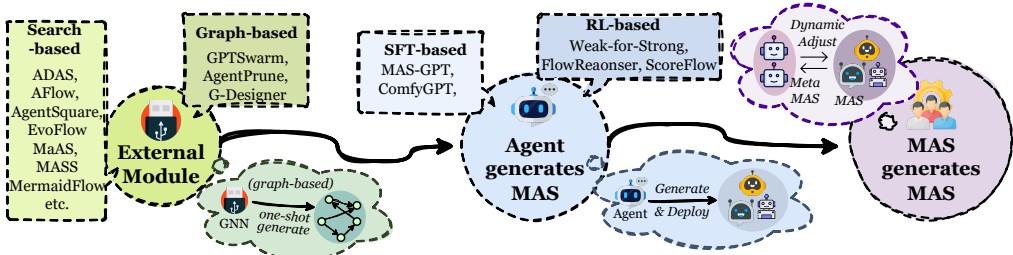

Figure 1: The paradigm shift of automatic multi-agent system design: from external module-based MAS generation (*e.g.*, by GNNs, evolutionary algorithms, and search algorithms) and agent-based generation (*i.e.*, FlowReasoner (Gao et al., 2025) and MAS-GPT (Ye et al., 2025b)) to $\mathrm{MAS}^2$, where a MAS recursively generates another MAS.

alike, as exemplified by ADAS (Hu et al., 2025b), MaAS (Zhang et al., 2025b), and AFlow (Zhang et al., 2024c). This shift has inaugurated a phase of rapid, genuinely transformative advance.

Nevertheless, not all autonomously constructed MAS are alike. Early approaches often relied on **external modules** such as Bayesian optimization (Li et al., 2023b), Monte Carlo tree search (MCTS) (Zhang et al., 2024c; Liang et al., 2025), or graph neural networks (Zhang et al., 2024b). While effective to some extent, these methods were confined to a predefined search space of atomic operators (*e.g.*, Chain-of-Thought (CoT), Reflexion, Debate), limiting their capacity for architectural innovation. More recently, researchers have shifted toward enabling **agents themselves** to construct MAS, employing techniques such as SFT (Ye et al., 2025b), DPO (Wang et al., 2025b), and GRPO (Nie et al., 2025). These advances allow task-level adaptivity, whereby distinct systems can be customized for each problem instance. Yet most of these methods still adhere to a "*generate-once-and-deploy*" paradigm, where a system is instantiated once per task and executed unchanged (Zhang et al., 2025b), regardless of success or failure. This paradigm proves fundamentally inadequate, as real-world interactions between multi-agent systems and their environments are inherently dynamic and error-prone (*e.g.*, network failures, tool crashes, file loss) (Cemri et al., 2025). Consequently, "*generate-once-and-deploy*" strategies are highly susceptible to collapse from a single unforeseen disruption and lack the capacity to adapt beyond their initial instantiation.

Having surveyed the evolution and inherent limitations of both external module-based and agent-driven automatic agent systems, we propose a third new paradigm that inherently enables self-generation and self-adaptation of MAS. Specifically, we envision a framework in which **a multi-agent system autonomously constructs another multi-agent system**, which we refer to as $\mathrm{MAS}^2$. Unlike previous approaches, $\mathrm{MAS}^2$ orchestrates a specialized, LLM-facilitated meta-agent team tasked with *generating*, *configuring*, and *rectifying* systems in a task-adaptive and progress-aware manner. By internalizing distinct construction responsibilities across dedicated meta-agents, $\mathrm{MAS}^2$ transcends the creative constraints of external modules and overcomes the rigidity of traditional "generate-once-and-deploy" strategies.

In this paper, we introduce a meta multi-agent system (meta MAS) designed to instantiate our proposed $\mathrm{MAS}^2$ paradigm. This system is built upon a tri-agent architecture comprising a ♣ *generator*, an ¶ *implementor*, and a ♠ *rectifier*, each undergoing specialized training to internalize its distinct meta-generative function. Specifically, for any given query, the *generator* agent architects a high-level, multi-agent workflow template, which outlines the sequence of agentic operations. Subsequently, the *implementor* agent instantiates this template by populating each procedural step with a concrete LLM backbone, rendering the workflow fully executable. During runtime, the *rectifier* agent actively monitors the execution state and environmental feedback, issuing timely corrections to the system for adaptiveness to dynamic conditions.

To facilitate training and data efficiency, we devise an offline reinforcement learning (RL) strategy for trajectory collection and optimization, termed **Collaborative Tree Optimization (CTO)**. Within the CTO framework, the three agents collaboratively expand a decision tree representing diverse MAS configurations and execution pathways. Upon reaching a terminal state, a final environmental feedback signal is obtained. We then employ a *path credit propagation* mechanism to attribute the final outcome to the upstream decisions made by each agent. This process yields role-specific preference data, which is then leveraged through relative reward-based preference alignment algorithms to effectively specialize the distinct meta-generative function of each agent. In this way, $\mathrm{MAS}^2$ mate-

rializes a fully autonomous and self-adaptive multi-agent system pipeline, affording robustness and precision commensurate with the demands of intricate, long-horizon task.

In brief, our contribution can be summarized as follows:

❶ **Paradigm Formulation**: We introduce the $\texttt{MAS}^2$ paradigm, wherein a multi-agent system itself is employed to generate diverse multi-agent systems, thereby achieving greater creativity and task adaptiveness than previous external module or agent-based generation pipelines.

❷ **Methodology Proposal**: We instantiate $\texttt{MAS}^2$ through a "generator–implementor–rectifier" tri-agent framework, and design a collaborative tree optimization (CTO) procedure to enable efficient specialization and training of meta-agents.

❸ **Empirical Evaluation**: Comprehensive experiments across six benchmarks demonstrate that $\texttt{MAS}^2$ achieves **(I) superior competence**, surpassing state-of-the-art multi-agent systems such as MaAS and ScoreFlow by up to $9.3\%$; **(II) Pareto-optimal cost–performance**, achieving the highest performance along the cost–performance frontier; and **(III) cross-backbone generalization**, effectively leveraging unseen LLMs to deliver improvements of up to $15.1\%$.

## 2 RELATED WORK

**Automating Agent Systems**  concerns multi-agent systems whose configurations (including agent prompting, communication protocols, tool integration, and LLM backbone selection) are automatically determined without substantial human intervention. Early efforts primarily relied on external control modules: for instance, GPTSwarm (Zhuge et al., 2024) and AgentPrune (Zhang et al., 2024a) employed parameterized adjacency matrices; G-Designer (Zhang et al., 2024b) and GraphRouter (Feng et al., 2024) utilized graph neural networks; while MasRouter (Yue et al., 2025) adopted a variational autoencoder. A broader class of systems, such as ADAS (Hu et al., 2024a), AgentSquare (Shang et al., 2024), AFlow (Zhang et al., 2024c), MaAS (Zhang et al., 2025b), MASS (Zhou et al., 2025), and MermaidFlow (Zheng et al., 2025), similarly follow this paradigm. However, the generative capacity of external modules proved inherently constrained, giving rise to model-driven approaches for constructing MAS. MAS-GPT (Ye et al., 2025b) employs large-scale query–workflow pairs for supervised fine-tuning of a system-generating agent, while Score-Flow (Wang et al., 2025b) adopts a similar strategy under DPO. FlowReasoner (Gao et al., 2025) and Weak-for-Strong (Nie et al., 2025), leverage GRPO-style online RL (Guo et al., 2025) to train meta-agents for MAS generation. In contrast, our proposed $\texttt{MAS}^2$ transcends the limitations of the "generate-once-and-deploy" paradigm by enabling self-adjusting and resilient construction of MAS.

**Meta LLM Agents.**  Meta agents typically denote LLM-based coordinating entities that provide meta-level guidance or generation for agent teams (Xiong et al., 2025). Most existing meta agents are instantiated directly from off-the-shelf, powerful LLMs and are pervasive across domains. For instance, in deep research systems, Camel's OWL (Hu et al., 2025a), Skywork's AgentOrchestra (Zhang et al., 2025d), ByteDance's AIME (Shi et al., 2025), Tencent's Cognitive Kernel-Pro (Fang et al., 2025), and several other frameworks (Bahdanau et al., 2024) all rely on a meta agent to allocate and regulate tasks among sub-agents. Analogously, in software engineering systems, manager or "CEO" agents are widely adopted (Hong et al., 2023; Qian et al., 2023; Hu et al., 2024b). More recently, a smaller body of work has explored explicitly training a meta agent as a leader (Estornell et al., 2025; Gao et al., 2025). Distinct from these efforts, $\texttt{MAS}^2$ elevates the concept further by transforming the meta agent from a single controlling entity into a meta-MAS.

**RL for MAS.**  RL has been witnessed to propel the agentization of LLMs at an unprecedented pace (Zhang et al., 2025a), substantially advancing their capabilities in reasoning, tool utilization, memory management, and beyond. Within the specific context of multi-agent systems (MAS), existing research on RL for MAS can be broadly categorized into three streams: *(i) training external modules*, which leave the internal parameters of agents untouched while exclusively training external control modules via RL (Zhang et al., 2025b; Wang et al., 2025a;d); *(ii) training partial agents*, which selectively train a subset of agents while keeping the remainder fixed, as exemplified by MLPO (Estornell et al., 2025); and *(iii) comprehensively training all agents* that jointly update and evolve all constituent agents, such as Sirius (Zhao et al., 2025), MALT (Motwani et al., 2024), MaPoRL (Park et al., 2025), MARFT (Liao et al., 2025), among others (Wan et al., 2025; Wang et al., 2025c; Xia et al., 2025; Liu et al., 2025).

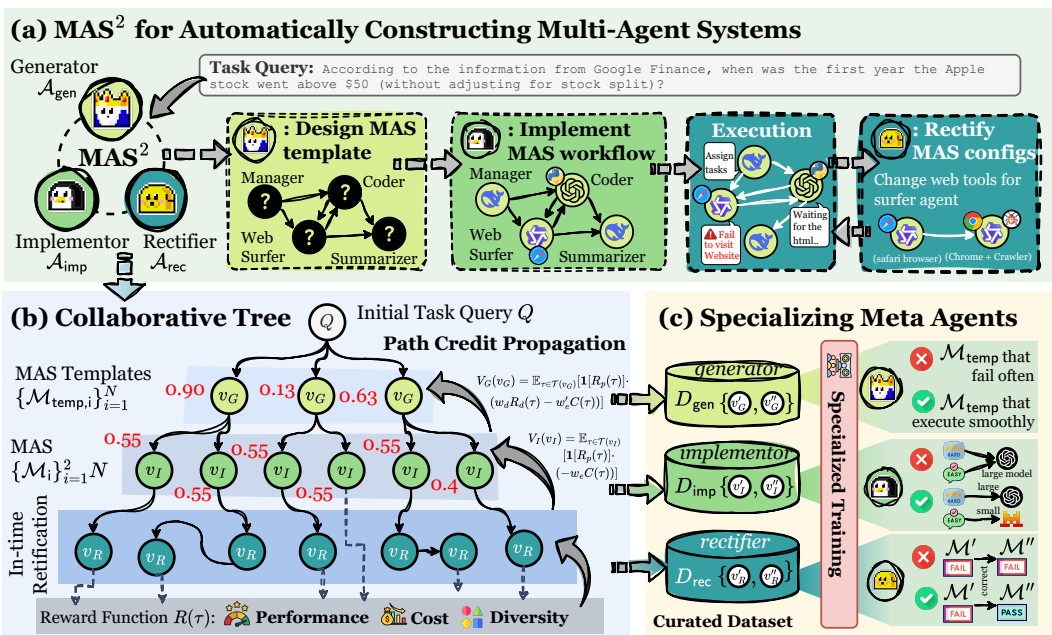

Figure 2: The framework overview of our proposed MAS$^2$.

## 3 METHODOLOGY

Figure 2 presents the overall workflow of MAS$^2$. At inference time, MAS$^2$ accepts each task query as input: the *generator* agent produces a MAS template, the *implementer* instantiates its components, and the *rectifier* continuously monitors and adapts execution in real time ($\triangleright$ Section 3.1). These meta-agents are trained under our *collaborative tree optimization* (CTO) framework, which collects preference signals through path-level credit propagation ($\triangleright$ Section 3.2), and subsequently leverages them for targeted meta-agent training ($\triangleright$ Section 3.3).

### 3.1 META-SCHEDULING OF MULTI-AGENT SYSTEMS

The effective deployment of an MAS for a novel, complex task necessitates a departure from static, pre-defined architectures. Real-world problems are inherently dynamic, demanding a framework capable of composing and adapting an MAS on-the-fly. Therefore, we introduce our MAS$^2$ to dynamically construct a bespoke MAS tailored to the specific demands of a given task query, $Q$. We begin by formalizing the constitution of a target MAS $\mathcal{M}$ as four core operational components:

$$\mathcal{M} = \langle \mathcal{R}, \mathcal{P}, \mathcal{T}, \mathcal{B} \rangle, \tag{1}$$

where $\mathcal{R} = \{r_1, \ldots, r_{|\mathcal{R}|}\}$ denotes a set of allocated agents, $\mathcal{P} = \{\rho_{ij} \mid r_i, r_j \in \mathcal{R}\}$ defines the communication protocol by specifying the permissible message structures $\rho_{ij}$ between agents, $\mathcal{T} = \{t_1, \ldots, t_{|\mathcal{T}|}\}$ is a set of available tools (*e.g.*, python interpreter, playwright browser), and $\mathcal{B} = \{b_i \mapsto r_i\}_{r_i \in \mathcal{R}}$ assigns concrete LLM backbones $b_i$ to each agent.

**Generator.** The process commences with the *generator* agent $\mathcal{A}_{\text{gen}}$, which acts as the system architect. Given the query $Q$, the generator formulates a strategic blueprint for the MAS by producing a workflow template, $\mathcal{M}_{\text{temp}}$, which abstracts away the final computational resources:

$$\mathcal{M}_{\text{temp}} = \langle \mathcal{R}, \mathcal{P}, \mathcal{T} \rangle \sim \pi_{\text{gen}}(\cdot | Q), \tag{2}$$

where $\pi_{\text{gen}}$ is the generator's policy. Equation (2) provides a complete yet uninstantiated MAS plan.

**Implementor.** Next, the *implementor* agent $\mathcal{A}_{\text{imp}}$ translates the template $\mathcal{M}_{\text{temp}}$ into a fully executable system. It defines an assignment policy, $\phi$, that maps each role to a concrete LLM backbone from a candidate LLM pool $\mathbb{L} = \{b_1, \ldots, b_{|\mathbb{L}|}\}$:

$$\phi : \mathcal{R} \to \mathbb{L}, \quad r_i \mapsto b_{j(i)} \quad \text{where } \phi \sim \pi_{\text{imp}}(\cdot | \mathcal{M}_{\text{temp}}, \mathbb{L}, Q), \tag{3}$$

where $\pi_{\mathsf{imp}}$ is the implementor's policy and $j(i)$ indexes the selected backbone for agent $r_i$. The final MAS is then composed as

$$\mathcal{M} = \mathcal{M}_{\mathsf{temp}} \oplus \{(r_i, \phi(r_i)) \mid r_i \in \mathcal{R}\}, \tag{4}$$

where $\oplus$ denotes the composition that integrates the backbone assignments into the workflow template, producing a fully instantiated, executable MAS. We provide illustrative examples of the multi-agent systems generated by $\mathrm{MAS}^2$ in Appendix C.

**Rectifier.** With the MAS $\mathcal{M}$ instantiated, it is deployed to begin execution. The *rectifier* agent $\mathcal{A}_{\mathsf{rec}}$ assumes an online monitoring role to ensure system resilience. The agent is invoked if its trigger function $A_R$, which monitors the execution state $s_t$, evaluates to 1:

$$A_R(s_t) = \mathbb{1}\left[C(s_t) > \theta_C \vee O(s_t) = \mathtt{Failure}\right], \tag{5}$$

where $\mathbb{1}[\cdot]$ is the indicator function. The rectifier is triggered if either the cumulative resource consumption $C(s_t)$ (*e.g.*, token count, execution steps) exceeds a budget $\theta_C$, or if the operational outcome $O(s_t)$ results in an explicit failure (*e.g.*, search engine failure, code execution error). Upon activation, the rectifier intervenes by generating a modification to the current system configuration, $\mathcal{M}_t$. This can range from local adjustments (*e.g.*, re-assign proper tools, modify agent prompts) to global architectural changes (*e.g.*, revise workflow codes). The rectifier's intervention results in:

$$\mathcal{M}_{t+1} \sim \pi_{\mathsf{rec}}(\cdot | \mathcal{M}_t, s_t), \tag{6}$$

where $\pi_{\mathsf{rec}}$ is the rectifier's learned policy. The system then resumes execution from state $s_t$ using the updated configuration $\mathcal{M}_{t+1}$. We give illustrative examples of how the rectifier modifies an erroneous MAS in Appendix D. This synergistic process ensures that the generated MAS is not only custom-built for the task but also possesses the capacity for real-time self-correction, a key advantage for tackling long-horizon, unpredictable problem domains.

## 3.2 TRAINING TRAJECTORY CURATION

To enable the specialization of our meta-agents, we introduce our **Collaborative Tree Optimization (CTO)** framework to construct a collaborative decision tree, which serves as the formal basis for our data curation. Formally, we represent the structure as a rooted, directed tree $\mathcal{G}_Q = (V, E)$ associated with a task query $Q$. The vertex set $V$ contains the query root node $v_Q$, a layer of generator nodes $\{v_G\}$, a layer of implementor nodes $\{v_I\}$, rectifier nodes $\{v_R\}$ that may appear adaptively during execution, and terminal leaf nodes $\{v_F\}$. The tree is populated through a sequential sampling process: the generator branches out $K$ candidate templates from the root, the implementor expands each with $N$ executable instantiations, and the rectifier may introduce further branches by adjusting the current MAS. A trajectory $\tau$ is defined as a unique path from the root $v_Q$ to a terminal node $v_F$:

$$\tau = (v_Q \xrightarrow{\mathcal{A}_{\mathsf{gen}}} v_G \xrightarrow{\mathcal{A}_{\mathsf{imp}}} v_I \ldots \xrightarrow{\mathcal{A}_{\mathsf{rec}}} v_F). \tag{7}$$

To evaluate such trajectories, a performance-only metric is inadequate, as it ignores the resource efficiency of the constructed MAS. We therefore introduce a conditional, cost-sensitive reward function $R(\tau)$, which assigns zero reward to failed trajectories and evaluates successful ones by their normalized resource consumption:

$$R(\tau) = \mathbb{1}[R_p(\tau)] \cdot \frac{1}{C_{\mathsf{norm}}(\tau)}, \ C_{\mathsf{norm}}(\tau) = \frac{C(\tau)}{\frac{1}{|\mathcal{T}|} \sum_{\tau' \in \mathcal{T}} C(\tau')}, \tag{8}$$

where $R_p(\tau) \in \{0, 1\}$ denotes the success status of trajectory $\tau$, and $C(\tau)$ measures raw resource consumption (*i.e.*, LLM API cost) and the denominator rescales costs relative to the empirical mean across the trajectory set $\mathcal{T}$. After assigning terminal rewards to the leaves of $\mathcal{G}_Q$, we propagate credit backward to intermediate decision nodes via **path credit propagation**. The value of a node $v \in V$ is defined as the expected reward of all trajectories that pass through it:

$$V(v) = \mathbb{E}_{\tau \in \mathcal{T}(v)}[R(\tau)] \approx \frac{1}{|\mathcal{T}(v)|} \sum_{\tau \in \mathcal{T}(v)} R(\tau), \tag{9}$$

where $\mathcal{T}(v)$ denotes the set of trajectories traversing $v$. This Monte Carlo estimate attributes terminal outcomes to upstream decisions, thereby providing each node with a principled value signal. The resulting annotated tree $\mathcal{G}_Q$ forms a structured dataset of preference-labeled decisions, which is subsequently used to train our meta-agents.

## 3.3 TRAINING AND SPECIALIZING META AGENTS

With the collaborative decision tree $\mathcal{G}_Q$ fully annotated with value estimates from the above curation process, the final objective is to specialize the meta-agent policies. A naive preference alignment treating all winning actions equally would discard the rich, quantitative information captured by our node values. To leverage this, following previous relative reward-based preference methods (Gao et al., 2024; Wang et al., 2025b), we leverage a value-guided optimization process to guide the training of the meta agents within MAS$^2$.

**Value-Guided Preference Construction.** First, we translate the value-annotated tree into preference data. For each non-terminal node $v \in \mathcal{G}_Q$ representing a decision context $c_v$, we form preference tuples. Unlike standard binary preference pairs, our tuples incorporate the quantitative margin of victory:

$$\mathcal{D}_\pi = \left\{ (c_v, a_{\text{win}}, a_{\text{lose}}, \Delta V) \middle| \begin{matrix} (v, a_{\text{win}}), (v, a_{\text{lose}}) \in E, \\ \Delta V = V(v') - V(v'') > 0 \end{matrix} \right\} \tag{10}$$

where $v'$ and $v''$ are the successor nodes of $a_{\text{win}}$ and $a_{\text{lose}}$ respectively. The term $\Delta V$ represents the "preference strength," a crucial signal indicating how much better one decision was than another.

**Policy Specialization via Value-Scaled Optimization.** Our training process refines a reference policy $\pi_{\text{ref}}$ (the vanilla LLM) into a specialized policy $\pi^*$ (a specialized generator/implementer/rectifier). Our loss function is designed to incorporate the preference strength $\Delta V$ to modulate the gradient during training. This ensures that the model learns most from high-confidence preference pairs where the value difference is significant, while treating low-margin pairs as less influential. The optimization objective for each policy $\pi_\theta$ is to minimize the following value-scaled loss:

$$\mathcal{L}_{\text{CTO}}(\pi_\theta; \pi_{\text{ref}}) = -\mathbb{E}_{(c, a_{\text{win}}, a_{\text{lose}}, \Delta V) \sim \mathcal{D}_\pi} \left[ \Delta V \cdot \log \sigma \left( \beta \log \frac{\pi_\theta(a_{\text{win}}|c)}{\pi_{\text{ref}}(a_{\text{win}}|c)} - \beta \log \frac{\pi_\theta(a_{\text{lose}}|c)}{\pi_{\text{ref}}(a_{\text{lose}}|c)} \right) \right] \tag{11}$$

where $\sigma(\cdot)$ is the logistic function and $\beta$ is a temperature parameter. By weighting each term in the expectation by $\Delta V$, our loss function directly prioritizes learning from the most unambiguous and impactful decisions identified during trajectory curation.

This optimization procedure is applied independently to the generator, implementor, and rectifier, yielding three specialized policies, $\pi^*_{\text{gen}}$, $\pi^*_{\text{imp}}$, and $\pi^*_{\text{rec}}$. Together, these components establish a unified methodology that not only enables the dynamic construction of bespoke multi-agent systems, but also ensures their continual refinement through curated training and value-guided specialization.

## 4 EXPERIMENTS

### 4.1 EXPERIMENTAL SETUP

**Benchmarks.** To provide a comprehensive evaluation of MAS$^2$, we employ eight benchmarks spanning four domains: ■ **multi-hop search**, including HotpotQA (Yang et al., 2018), Bamboogle (Press et al., 2022), and Natural Question (NQ) (Kwiatkowski et al., 2019); ■ **deep research**, represented by BrowseComp+ (Chen et al., 2025b), an enhanced variant of OpenAI's BrowseComp (Wei et al., 2025) that corrects erroneous cases and enables more stable offline evaluation; ■ **code generation**, comprising HumanEval (Chen et al., 2021) and MBPP (Austin et al., 2021); and ■ **mathematical reasoning**, assessed with MATH (Hendrycks et al., 2021).

**Baselines.** The baselines against which we compare MAS$^2$ can be broadly grouped into three categories: ■ **vanilla LLMs**, where single models are directly evaluated, alongside prompting strategies such as CoT (Wei et al., 2023) and Self-Consistency (Wang et al., 2023a); ■ **handcrafted multi-agent systems**, including MedPrompt (Nori et al., 2023), MultiPersona (Wang et al., 2023b), LLM-Debate (Du et al., 2023), and DyLAN (Liu et al., 2024); ■ **(partially) automated multi-agent systems**, comprising ADAS (Hu et al., 2025b), MaAS (Zhang et al., 2025c), AFlow (Zhang et al., 2024c), and ScoreFlow (Wang et al., 2025b). For multi-agent system baselines, we instantiate them using the relatively highest-performing QwQ-32B and GPT-4o and report the average scores.

**Parameter & Model Configurations.** The LLM pool available for MAS$^2$ is specified as GPT-4o and GPT-4o-mini (Liu et al., 2024), Qwen2.5-72b-instruct (Qwen et al., 2025), Qwen3-14b (Yang

Table 1: **Performance comparison** across 8 benchmarks across 13 baselines. Each cell reports the average of **three random runs**. The best/second-best results are **bolded**/underlined.

| Model | Multi-hop Search | | | Deep Research | Code Generation | | Math |
|---|---|---|---|---|---|---|---|
| | HotpotQA | Bamboogle | NQ | BrowseComp+ | HumanEval | MBPP | MATH |
| Qwen3-14B | 62.2 | 34.4 | 51.9 | 13.1 | 75.6 | 40.8 | 61.5 |
| GPT-4o-mini | 64.1 | 26.4 | 70.7 | 13.7 | 87.8 | 46.0 | 51.2 |
| QwQ-32B | 64.9 | 39.2 | 61.5 | 2.8 | 51.8 | 82.2 | 65.6 |
| Qwen-2.5-72B | 66.8 | 31.2 | 63.0 | 2.6 | 82.3 | 76.0 | 55.3 |
| GPT-4o | 69.5 | 49.6 | 71.1 | 13.2 | 89.6 | 73.4 | 56.5 |
| Avg. (Above) | 65.5 | 36.2 | 63.6 | 9.5 | 77.4 | 63.7 | 58.0 |
| CoT (GPT-4o) | $66.2_{\uparrow 0.7}$ | $52.8_{\uparrow 16.6}$ | $68.7_{\uparrow 5.1}$ | $\underline{16.1}_{\uparrow 7.6}$ | $90.8_{\uparrow 13.4}$ | $75.4_{\uparrow 11.7}$ | $58.9_{\uparrow 0.9}$ |
| SC (GPT-4o) | $66.4_{\uparrow 0.9}$ | $54.4_{\uparrow 18.2}$ | $73.6_{\uparrow 10.0}$ | $11.8_{\uparrow 2.3}$ | $\underline{96.3}_{\uparrow 18.9}$ | $75.1_{\uparrow 11.4}$ | $62.0_{\uparrow 4.0}$ |
| MedPrompt | $72.4_{\uparrow 6.9}$ | $48.0_{\uparrow 11.8}$ | $66.1_{\uparrow 2.5}$ | $12.0_{\uparrow 2.5}$ | $92.1_{\uparrow 14.7}$ | $69.2_{\uparrow 5.5}$ | $59.7_{\uparrow 1.7}$ |
| MultiPersona | $71.1_{\uparrow 5.6}$ | $50.2_{\uparrow 14.0}$ | $68.4_{\uparrow 4.8}$ | $10.1_{\uparrow 0.6}$ | $92.9_{\uparrow 15.5}$ | $70.4_{\uparrow 6.7}$ | $53.7_{\downarrow 4.3}$ |
| LLM-Debate | $66.9_{\uparrow 1.4}$ | $54.4_{\uparrow 18.2}$ | $70.8_{\uparrow 7.2}$ | $15.3_{\uparrow 5.8}$ | $88.7_{\uparrow 11.3}$ | $70.3_{\uparrow 6.6}$ | $67.3_{\uparrow 9.3}$ |
| DyLAN | $80.8_{\uparrow 15.3}$ | $59.7_{\uparrow 23.5}$ | $72.1_{\uparrow 8.5}$ | $15.8_{\uparrow 6.3}$ | $90.4_{\uparrow 13.0}$ | $77.3_{\uparrow 13.6}$ | $65.7_{\uparrow 7.7}$ |
| ADAS | $78.5_{\uparrow 13.0}$ | $50.8_{\uparrow 14.6}$ | $65.9_{\uparrow 2.3}$ | $7.0_{\downarrow 2.5}$ | $88.8_{\uparrow 11.4}$ | $68.7_{\uparrow 5.0}$ | $51.7_{\downarrow 6.3}$ |
| MaAS | $83.6_{\uparrow 18.1}$ | $62.0_{\uparrow 25.8}$ | $76.0_{\uparrow 12.4}$ | $14.0_{\uparrow 4.5}$ | $92.8_{\uparrow 15.4}$ | $82.2_{\uparrow 18.5}$ | $70.1_{\uparrow 12.1}$ |
| AFlow | $77.9_{\uparrow 12.4}$ | $59.2_{\uparrow 23.0}$ | $74.5_{\uparrow 10.9}$ | $10.0_{\uparrow 0.5}$ | $92.9_{\uparrow 15.5}$ | $82.9_{\uparrow 19.2}$ | $68.5_{\uparrow 10.5}$ |
| ScoreFlow | $\underline{86.0}_{\uparrow 20.5}$ | $\underline{64.8}_{\uparrow 28.6}$ | $\underline{76.4}_{\uparrow 12.8}$ | $10.4_{\uparrow 0.9}$ | $95.9_{\uparrow 18.5}$ | $\underline{84.7}_{\uparrow 21.0}$ | $64.4_{\uparrow 6.4}$ |
| $\text{MAS}^2$ | $\mathbf{89.3}_{\uparrow 23.8}$ | $\mathbf{67.2}_{\uparrow 31.0}$ | $\mathbf{79.1}_{\uparrow 15.5}$ | $\mathbf{19.7}_{\uparrow 10.2}$ | $\mathbf{97.0}_{\uparrow 19.6}$ | $\mathbf{85.1}_{\uparrow 21.4}$ | $\mathbf{71.3}_{\uparrow 13.3}$ |

et al., 2025b), and QwQ-32B (Team, 2025). We access the above models via the OpenRouter [1] API. The backbone LLM for the generator, implementer, and rectifier in $\text{MAS}^2$ are consistently set as Qwen3-8B (Yang et al., 2025b). For data curation in Section 3.2, the generator expands into four candidate $v_G$ nodes, and each $v_G$ is further instantiated twice by the implementor, yielding two corresponding $v_I$ nodes per branch. In the RL training in Section 3.3, we set $\texttt{epoch\_num} = 2$, $\texttt{learning\_rate} = 5 \times 10^{-5}$, and $\texttt{gradient\_accumulation\_steps} = 4$. We finetune the base Qwen3-8B via LoRA, with $\texttt{rank} = 8$ and $\texttt{alpha} = 16$.

## 4.2 MAIN RESULTS

Table 1 compares $\text{MAS}^2$ against all single-LLM baselines within the LLM pool, as well as eight representative MAS baselines, from which we distill the following key observations:

**Obs. ❶: Existing automated MAS fail to generalize across domains.** Table 1 shows that current MAS baselines perform inconsistently across tasks. For example, MultiPersona performs well on HumanEval (92.9%) and Bamboogle (50.2%) yet drops on MATH ($-4.3\%$) compared with single LLMs. LLM-Debate excels on Math (67.3%) and Bamboogle (54.4%) but lacks BrowseComp+ coverage. Stronger baselines like ADAS and MaAS show mixed performance: ADAS scores 78.5% on HotpotQA and 88.8 on HumanEval but falls on deep research (merely 7.0% on BrowseComp+) and MATH (51.7%), while MaAS is strong on multi-hop search (83.6% on HotpotQA) and code generation (92.8% on HumanEval) yet remains weak on BrowseComp+ (14.0%). Overall, these results indicate that existing MASs often excel in some domains but fail to generalize consistently.

**Obs. ❷: $\text{MAS}^2$ outperforms both handcrafted and automated multi-agent systems.** (I) Compared to handcrafted MAS, $\text{MAS}^2$ obtains SOTA performance across multiple domains, such as mathematics, code, and deep research. Concretely, $\text{MAS}^2$ achieves 89.3% on HotpotQA, surpassing the best manual setting DyLAN by 8.5%, and outperforming MedPrompt by 16.9%. In MATH, $\text{MAS}^2$ scores 71.3%, slightly outperforming LLM-Debate by 4.0%. (II) $\text{MAS}^2$ also excels when compared to automated MAS. In the multi-hop search tasks (HotpotQA, Bamboogle, NQ), $\text{MAS}^2$ achieves an average of $\sim 78.5\%$, surpassing ScoreFlow and MaAS by 2.8% and 4.7%, respectively. In code generation, $\text{MAS}^2$ scores 97.0% on HumanEval, outperforming AFlow by 4.1%. In the MATH task, $\text{MAS}^2$ achieves 71.3%, improving by 6.9% over ScoreFlow and 2.8% over AFlow. In summary, $\text{MAS}^2$ consistently demonstrates strong performance across multiple task domains, validating its robust multi-task adaptability and effectiveness.

---

[1]https://openrouter.ai/

### 4.3 GENEALIZATION STUDY

Although MAS$^2$ was trained using a fixed LLM pool, including models such as GPT-4o, we aim to demonstrate that its capabilities extend beyond previously seen LLMs and can effectively leverage unseen models. To evaluate this, we augment the LLM pool at inference with three stronger, previously unseen LLMs (Qwen3-Coder (Yang et al., 2025a), GPT-5-Mini, and Gemini-2.5-Pro (Comanici et al., 2025)) while retaining the original models.

**Obs. ❸: MAS$^2$ generalizes to and exploits previously unseen LLM backbones.** Table 2 shows that MAS$^2$ effectively integrates these unseen LLM backbones, significantly improving performance. For instance, using the vanilla Qwen3-Coder, performance on MATH is 69.7% (cost $0.54) and 32.8% on Bamboogle (cost $0.06). When MAS$^2$ integrates Qwen3-Coder and GPT-5-Mini, MATH performance improves to 88.6% (cost $2.22) and Bamboogle to 82.5% (cost $0.53). Further integration with GPT-5-Mini and Gemini-2.5-Pro boosts MATH performance to 90.6% (cost $16.14) and Bamboogle to 84.0% (cost $2.73), offering significant gains

Table 2: Performance results when integrated with stronger unseen LLMs. "Vanilla LLM" denotes merely using one LLM backbone for testing. Checkmarks are used to denote the LLMs used.

| Method | Stronger LLMs | | | Datasets | | | |
| | Qwen3-Coder | GPT-5-Mini | Gemini-2.5-Pro | MATH | | Bamboogle | |
| | | | | Perf.(%) | Cost($) | Perf.(%) | Cost($) |
|---|---|---|---|---|---|---|---|
| Vanilla LLM | ✔ | | | 69.7 | 0.54 | 32.8 | 0.06 |
| | | | ✔ | 68.0 | 14.48 | 83.2 | 2.27 |
| | | ✔ | | 87.3 | 1.22 | 80.8 | 0.21 |
| MAS$^2$ | ✘ | ✘ | ✘ | 71.3 | 0.74 | 67.2 | 0.15 |
| | ✔ | ✔ | ✘ | 88.6 | 2.22 | 82.5 | 0.53 |
| | ✔ | ✘ | ✔ | 84.8 | 15.47 | 84.2 | 2.57 |
| | ✘ | ✔ | ✔ | 90.6 | 16.14 | 84.0 | 2.73 |

over the vanilla LLM baseline while maintaining a reasonable cost increase. These results confirm that MAS$^2$ not only generalizes to new LLM backbones but also harnesses their strengths to achieve higher accuracy, without requiring additional fine-tuning. This validates the flexibility of MAS$^2$ for practical deployment.

### 4.4 COST ANALYSIS

To evaluate the practical viability of our method, we analyze its economic efficiency by exploring the performance-cost trade-off. As shown in Figure 3, we compare MAS$^2$ against a wide range of baselines to demonstrate its ability to achieve superior results without incurring prohibitive costs.

**Obs. ❹: MAS$^2$ establishes a new cost-performance Pareto frontier.** Figure 3 demonstrates that MAS$^2$ (indicated by the ★ symbol) consistently achieves a state-of-the-art balance, delivering higher accuracy than methods of similar cost (*e.g.*, ScoreFlow), and significantly lower cost than methods with comparable accuracy. For

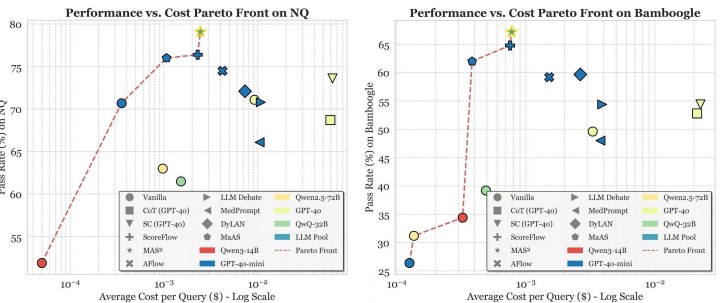

Figure 3: Cost-performance trade-off on the NQ and Bamboogle benchmark.

instance, on Bamboogle, MAS$^2$ achieves a 12.8% higher pass rate than the expensive SC (GPT-4o) baseline while being over 25× cheaper. Similarly, on NQ, it surpasses the strong ScoreFlow baseline by 2.7% in accuracy at a nearly identical cost, while simultaneously outperforming SC (GPT-4o) by 5.5 points and cutting costs by over 20 times. This Pareto efficiency mainly stems from MAS$^2$'s ability to operate dynamically at (1) the LLM level, assigning simpler tasks to smaller LLMs and reserving larger, more capable LLMs for complex reasoning, and also (2) system-level, configuring lightweight multi-agent systems for easy problems while deploying more sophisticated system architectures for challenging ones, achieving both accuracy and cost-effectiveness.

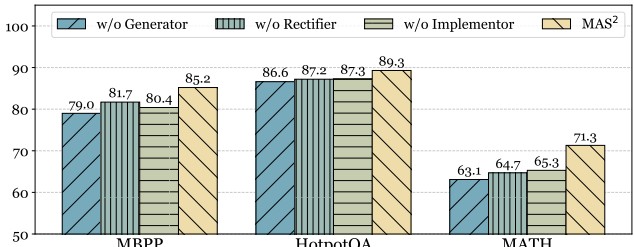

Figure 4: The ablation study of $\mathrm{MAS}^2$.

## 4.5 FRAMEWORK ANALYSIS

**Ablation Study.** We conduct ablation studies on MBPP, HotpotQA, and MATH to assess the contribution of each $\mathrm{MAS}^2$ component. Specifically, we consider three ablations: *w/o Generator* and *w/o Implementor* correspond to using the untrained Qwen3-8B model as the generator and implementor, respectively, while *w/o Rectifier* denotes the removal of the rectifier, which monitors and modifies the MAS dynamically. As shown in Figure 4, removing the generator (*w/o Generator*) results in substantial performance drops (from 85.2% to 79.0% on MBPP, from 89.3% to 86.6% on HotpotQA). Similarly, *w/o Implementor* also causes notable degradation, with scores of 80.4% on MBPP, 87.3% on HotpotQA, and 65.3% on MATH, demonstrating that the trained implementor agent effectively learns to allocate LLMs optimally for different tasks. *w/o Rectifier* leads to intermediate drops of 3.5% on MBPP and 6.6% on MATH, indicating that the rectifier is crucial for real-time self-adjustment during interactions with the environment. These results confirm that all three modules are indispensable, and omitting any component leads to clear performance degradation across multiple domains.

**Case Study** Figure 5 shows task-specific multi-agent systems designed by $\mathrm{MAS}^2$ that strategically allocate different LLMs, each with customized complexity and workflow configurations.

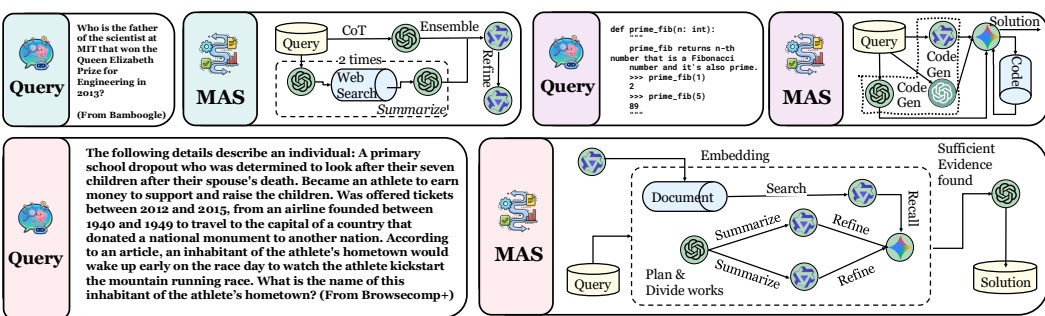

Figure 5: A visualization of task-specific multi-agent systems designed by $\mathrm{MAS}^2$ on Bamboogle, HumanEval, and BrowseComp+ benchmarks.

The explanation of each devised multi-agent system is as follows:

- **(I) Bamboogle (QA)**: In QA tasks, the designed MAS utilizes multiple models (e.g., Qwen series) to generate candidate answers from different perspectives. Subsequently, a premier model (e.g., Gemini-2.5-pro) takes on the critical role of evaluating these candidates to deliver a final verdict.
- **(II) HumanEval (Coding)**: For code generation, the MAS design embodies a division of labor. More economical, lightweight models are assigned to generate and test multiple code solutions in parallel. Finally, a top-tier model (e.g., GPT-4o) is responsible for reviewing these initial proposals and synthesizing a final, high-quality program.
- **(III) BrowseComp+ (Research)**: In the MAS designed for complex research tasks, cognitive roles are clearly demarcated. The most powerful model (e.g., GPT-4o) focuses on high-level cognitive activities such as reflective reasoning, strategic planning, and information synthesis. Meanwhile, more cost-effective models are employed to execute the mechanical, high-throughput tasks of searching and retrieving information.

Results showcase how $\mathrm{MAS}^2$ flexibly designs heterogeneous MAS tailored to domain-specific demands. In the QA scenario, various models generate diverse initial answers, while a flagship model like GPT-4o is reserved for the final, decisive judgment. In contrast, for code generation, economical models handle parallel generation and testing, with the results ultimately being refined and elevated

by a more powerful model. In the research tasks, $\mathtt{MAS}^2$ dynamically coordinates models of different tiers to handle retrieval, summarization, and reflective reasoning. It is noteworthy that the MAS presented here are generated solely by the generator and implementor agents. In Appendix D, we further illustrate potential errors encountered by these MAS and demonstrate how the rectifier dynamically corrects them, thereby validating the adaptive capabilities of $\mathtt{MAS}^2$.

## 5 CONCLUSION

This work advances the frontier of automatic MAS by introducing $\mathtt{MAS}^2$, a paradigm grounded in recursive self-generation. Unlike prevailing approaches that follow a static "generate-once-and-deploy" doctrine, $\mathtt{MAS}^2$ instantiates a meta-level tri-agent architecture, comprising generator, implementer, and rectifier agents, that dynamically composes and adaptively rectifies task-specific MAS in response to real-time demands. Through the proposed Collaborative Tree Optimization, these meta-agents are trained and specialized, enabling robust adaptability and heightened efficiency. Empirical studies across seven benchmarks demonstrate that $\mathtt{MAS}^2$ not only achieves consistent gains over state-of-the-art MAS but also scales synergistically with stronger LLM backbones.

## 6 ACKNOWLEDGMENT

This work is supported by the National Research Foundation, Singapore, and DSO National Laboratories under the AI Singapore Programme (AISG Award No. AISG2-RP-2020-019); and by the National Research Foundation, Singapore, and the Cyber Security Agency of Singapore under the National Cybersecurity R&D Programme (NCRP25-P04-TAICeN). This research is also part of the IN-CYPHER Programme and is supported by the National Research Foundation, Prime Minister's Office, Singapore, under its Campus for Research Excellence and Technological Enterprise (CREATE) Programme.

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

## A  UTILIZATION OF LARGE LANGUAGE MODELS

In the preparation of this manuscript, LLMs are utilized to refine the language and formatting of initial drafts, aid in literature review and data visualization, and facilitate information retrieval.

## B  PROMPT SET

This appendix details the system prompts used to instruct the three core meta-agents within the $\text{MAS}^2$ framework: the Generator, the Implementer, and the Rectifier.

---

### ⚙ Prompt for the Generator Agent

**Description:** This prompt instructs the **Generator** agent. Its objective is to design a high-level, model-agnostic multi-agent system (MAS) workflow template based on a given user query.

```
"""
You are an expert in designing multi-agent system (MAS) workflows. Your task is to
    generate a Python script that defines a workflow for solving a given problem. The
    workflow must be encapsulated within a class named 'Workflow'.

## Rules and Constraints:
1.  **Class Structure**: The entire workflow logic must be contained within a single
    Python class named 'Workflow'.
2.  **Output Format**: Your entire response must be enclosed within '<graph>' and '</
    graph>' XML tags. Do not include any text or explanations outside of these tags.
3.  **Core Logic**: The main execution logic must be within an 'async def run_workflow(
    self)' method inside the 'Workflow' class.
4.  **Operators**: You must use the predefined operators listed below to construct the
    workflow. Do not invent new operators.
5.  **LLM Placeholder**: For any operator that requires an LLM, use the string
    placeholder '"llm_symbol"'. Do not use specific model names like "gpt-4o".

## Available Operators:
You can use the following operators to build your workflow. They are available under
    the 'operator' module.

-   **'operator.Custom(llm_config, instruction, output_type)'**: A general-purpose
    operator that takes an instruction and produces text.
-   **'operator.Search(llm_config, query_instruction, context_instruction)'**: An
    operator designed to perform web searches.
-   **'operator.Programmer(llm_config, instruction)'**: A specialized operator for
    generating code.
-   **'operator.Reviewer(llm_config, instruction)'**: An operator for reviewing and
    critiquing code or text.
-   **'operator.ScEnsemble(llm_config, instruction, num_candidates)'**: An operator for
     generating multiple candidate solutions and selecting the best one (Self-
    Consistency).
-   **'operator.Tool(tool_name)'**: An operator for executing a specific tool, like a
    Python interpreter ('tool_name="python"').

## Example Workflow Structure:
```python
# All necessary imports are provided automatically.

class Workflow:
    def __init__(self, problem: str, **kwargs):
        self.problem = problem
        # Define your operators here
        self.planner = operator.Custom("llm_symbol", instruction="Create a plan.")
        self.coder = operator.Programmer("llm_symbol", instruction="Write the code.")

    async def run_workflow(self):
        # Implement the workflow logic here by calling the operators
        plan = await self.planner(self.problem)
        code = await self.coder(plan)
        return code
Now, based on these instructions, please generate a workflow for the following problem.

Problem:
"""

***
```

---

```
## ## **`END_PROMPT` (End of the Prompt)**

This part provides the final instruction to start generating the code.

```python
"""
<graph>
# Remember to start your Python code here, beginning with the class definition.
class Workflow:
"""
```

---

🚀  Prompt for the Implementer Agent

**Description:** This prompt is provided to the **Implementer** agent. It takes the abstract workflow template from the Generator and instantiates it into an executable system by assigning a specific LLM backbone to each agent role from a predefined pool of available models.

```
"""
You are an expert model selector for AI workflows.

## Goal
Your primary goal is to replace every string placeholder "llm_symbol" in the operator
    constructors with the most suitable model name for that operator.

**DO NOT** modify any function signatures or add any extra keyword arguments like `llm
    =` or `model=` to method calls.

**ONLY** change the first argument of operator constructors. For example: `operator.
    Custom("llm_symbol", ...)` should become `operator.Custom("gpt-4o-mini", ...)`.

## Available LLMs
Here are the available LLMs you can choose from:
- **gpt-4o-mini**
- **gpt-4o**
- **qwen/qwen-2.5-72b-instruct**
- **qwen/qwq-32b**

## Reference LLM Catalog
This catalog provides brief descriptions of each model to help you make an informed
    decision:
- **gpt-4o-mini**: GPT-4o Mini is a smaller, faster variant of OpenAI's GPT-4o
    multimodal model. It's optimized for lower latency and is best suited for
    lightweight tasks or when speed is prioritized over peak performance.
- **gpt-4o**: GPT-4o is OpenAI's flagship multimodal model, offering exceptional
    performance in complex reasoning, including mathematical proofs and multi-step
    derivations. Its long-context capabilities make it ideal for high-precision
    evaluations.
- **qwen/qwen-2.5-72b-instruct**: This is a 72-billion-parameter instruction-tuned
    model that excels in advanced mathematical reasoning, theorem verification, and
    long-form derivations. It's one of the most powerful open-source models for high-
    stakes reasoning tasks.
- **qwen/qwq-32b**: This is a medium-sized, reasoning-optimized model from Qwen that is
     strong at step-by-step multi-hop reasoning and QA.

## Operator Descriptions
Here's what each operator does:
- **Custom**: Generates detailed step-by-step analysis and reasoning for factual
    questions, potentially using provided context.
- **AnswerGenerate**: Directly generates concise final answers for factual QA tasks
    based on evidence and reasoning.
- **ScEnsemble**: Evaluates multiple answer candidates and selects the most accurate
    one for factual questions.
- **Review**: Critiques and refines solutions for factual questions, ensuring accuracy
    and completeness.

## Critical Instructions
- **Only replace "llm_symbol" strings**; do not change "operator." paths.
- **Keep the exact same operator paths** as in the input code.
- **Do not add any import statements** or module prefixes.
- Your output **must be valid Python code** that can be executed.
- Your final output **must only be code**, starting with `<graph>` and ending with `</
    graph>`.

## Workflow Code to Modify
<graph>
class Workflow:
    def __init__(self, problem) -> None:
```

```
            self.problem = problem
            self.custom = operator.Custom("llm_symbol", self.problem)
            self.sc_ensemble = operator.ScEnsemble("llm_symbol", self.problem)

    async def run_workflow(self):
        """
        This is a workflow graph for the NQ dataset.
        """
        # Analyze the question with context
        analysis = await self.custom(instruction="Can you analyze this question and
            provide relevant information from the context?")
        # Ensemble multiple reasoning approaches
        final_answer = await self.sc_ensemble(solutions=[analysis])

        return {"solution": final_answer}
</graph>
"""
```

🔧  Prompt for the Rectifier Agent

**Description:** This prompt activates the **Rectifier** agent when the executing MAS encounters a failure (e.g., code execution error, API timeout) or enters a non-productive state. The agent's task is to analyze the runtime context and error logs to propose a concrete modification to the workflow code.

```
"""
You are an expert AI workflow debugger. Your task is to analyze and fix a broken Python
    workflow script based on the provided error log.

## Goal
Identify the root cause of the error in the "Broken Workflow" and correct the code. The
    corrected code must adhere to the operator definitions provided in the "Workflow
    Template."

## Critical Instructions
1.  **Analyze the Error**: Carefully read the "Error Log" to understand why the
    workflow failed.
2.  **Consult the Template**: Use the "Workflow Template" as a strict reference for the
    correct operator usage, including method names and argument formats.
3.  **Correct the Code**: Modify **only** the necessary lines in the `run_workflow`
    method of the "Broken Workflow" to resolve the error. Do not change the `__init__`
    method or operator definitions.
4.  **Output Format**: Your final output must **only** be the complete, corrected
    Python code for the `Workflow` class, enclosed within `<graph>` and `</graph>` tags
    . Do not include any explanations, comments, or apologies.

---
### Workflow Template & Operator Guide

<graph>
class Workflow:
    def __init__(self, problem) -> None:
        self.problem = problem
        self.code_generate = operator.CustomCodeGenerate("llm_symbol", self.problem)
        self.sc_ensemble = operator.ScEnsemble("llm_symbol", self.problem)
        self.test = operator.Test("llm_symbol", self.problem)

    async def run_workflow(self):
        # ... implementation ...
        pass
</graph>

Here are the only operators you can use:
1.  **CustomCodeGenerate**: Generates code.
    - **Format**: `code_generate(instruction: str) -> str`
2.  **ScEnsemble**: Selects the best solution from a list.
    - **Format**: `sc_ensemble(solutions: List[str]) -> str`
3.  **Test**: Modifies a solution using test cases.
    - **Format**: `test(solution: str) -> str`
---
### Broken Workflow

{broken_workflow}

---
### Error Log

{error_log}
```

```
---
### Corrected Workflow
"""
```

## C  EXAMPLES OF GENERATED MULTI-AGENT SYSTEMS

This section presents 5 representative multi-agent systems (represented in code) generated by $\text{MAS}^2$ for tasks of varying complexity and domain.

**Workflow for Multi-hop Question Answering (HotpotQA)**

```python
class Workflow:
    def __init__(
        self,
        problem
    ) -> None:
        self.problem = problem
        self.custom = operator.Custom("qwen/qwen-2.5-72b-instruct", self.problem)
        self.sc_ensemble = operator.ScEnsemble("qwen/qwq-32b", self.problem)
        self.answer_generate = operator.AnswerGenerate("gpt-4o-mini", self.problem)
        self.review = operator.Review("gpt-4o-mini", self.problem)

    async def run_workflow(self):
        """
        This is a workflow graph.
        """
        # Generate multiple candidate solutions with different custom instructions
        instructions = [
            "Provide a detailed explanation and answer.",
            "Give a concise answer with reasoning.",
            "Explain from a scientific perspective."
        ]
        solutions = [await self.custom(instruction) for instruction in instructions]

        # Use ensemble to select the best solution
        best_solution = await self.sc_ensemble(solutions)

        # Review the best solution to improve it
        reviewed_solution = await self.review(best_solution)

        # Generate final answer based on reviewed solution
        final_answer = await self.answer_generate()

        return final_answer
```

**Workflow for Code Generation (HumanEval)**

```python
class Workflow:
    def __init__(
        self,
        problem
    ) -> None:
        self.problem = problem
        self.code_generate_1 = operator.CustomCodeGenerate("gpt-4o-mini", self.problem)
        self.code_generate_2 = operator.CustomCodeGenerate("qwen/qwen3-14b", self.
            problem)
        self.code_generate_3 = operator.CustomCodeGenerate("gpt-4o-mini", self.problem)
        self.sc_ensemble = operator.ScEnsemble("gpt-4o", self.problem)
        self.test = operator.Test("gpt-4o-mini", self.problem)

    async def run_workflow(self):
        """
        This is a workflow graph.
        """
        solution_list = []
        for _ in range(3):
```

```
            solution = await self.code_generate_1(instruction="Please␣analyze␣the␣
                problem␣step␣by␣step␣and␣generate␣the␣code␣solution.")
            solution_list.append(solution)

        ensembled_solution = await self.sc_ensemble(solutions=solution_list)

        tested_solution = await self.test(solution=tested_solution if 'tested_solution'
            in locals() else ensembled_solution)

        return tested_solution
```

## 🔍 Workflow for Deep Research (BrowseComp+)

```
class Workflow:
    def __init__(
        self,
        problem
    ) -> None:
        self.problem = problem
        self.custom = operator.Custom("gpt-4o", self.problem)
        self.search = operator.Search("gpt-4o-mini", self.problem)
        self.browser = operator.Browser("gpt-4o-mini", self.problem)
        self.answer_generate = operator.AnswerGenerate("gpt-4o", self.problem)

    async def run_workflow(self):
        search_history = set()
        collected_evidence = ""
        max_iterations = 3
        iteration = 0
        query = self.problem

        while iteration < max_iterations:
            iteration += 1
            # Reflection: plan and rewrite query to optimize search
            reflection_prompt = f"<think>Iteration␣{iteration}:␣Plan␣a␣precise␣search␣
                query␣to␣find␣the␣learning␣institution␣matching␣all␣given␣criteria.␣
                Avoid␣repeating␣previous␣queries:␣{list(search_history)}</think>"
            rewritten_query = await self.custom(instruction=reflection_prompt)
            rewritten_query = rewritten_query.strip()
            if not rewritten_query or rewritten_query in search_history:
                # fallback to original problem if rewriting fails or repeats
                rewritten_query = query
            search_history.add(rewritten_query)

            # Search step
            search_results = await self.search(query=rewritten_query, top_k=5)
            if not search_results:
                # No results found, break early
                break

            # Extract docids and browse for full content
            new_evidence = []
            for result in search_results:
                docid = result.get("docid", "")
                if docid and docid not in search_history:
                    content = await self.browser(docid=docid)
                    if content and str(content).strip():
                        new_evidence.append(str(content).strip())
                    search_history.add(docid)

            if not new_evidence:
                # No new evidence found, break loop
                break

            # Accumulate evidence
            collected_evidence += "\n\n".join(new_evidence) + "\n\n"

            # Reflection: check if sufficient evidence collected
            reflection_check = f"<think>Iteration␣{iteration}:␣Given␣the␣accumulated␣
                evidence,␣is␣it␣sufficient␣to␣answer␣the␣question?␣If␣yes,␣stop␣
                searching.␣If␣no,␣refine␣the␣query␣for␣next␣iteration.</think>\n<search
                >{collected_evidence}</search>"
            decision = await self.custom(instruction=reflection_check)
            decision_lower = decision.lower()
            if "yes" in decision_lower or "sufficient" in decision_lower or "stop" in
                decision_lower:
```

```
            break

        # Final answer generation
        if collected_evidence.strip():
            solution = await self.answer_generate(context=collected_evidence.strip())
        else:
            solution = "Information_Not_Found_in_Context"

        return solution
```

### 🔢 Workflow for Mathematical Reasoning (MATH)

```python
class Workflow:
    def __init__(
        self,
        problem
    ) -> None:
        self.problem = problem
        self.gid = None
        # IMPORTANT: Each operator MUST be initialized with a model placeholder string
        #     "gpt-4o-mini"
        self.custom1 = operator.Custom("gpt-4o-mini", self.gid, self.problem)
        self.custom2 = operator.Custom("gpt-5-mini", self.gid, self.problem)
        self.programmer = operator.Programmer("qwen/qwen3-coder", self.gid, self.
            problem)
        self.review = operator.Review("gpt-5-mini", self.gid, self.problem)
        self.sc_ensemble = operator.ScEnsemble("gpt-5-mini", self.gid, self.problem)

    async def run_workflow(self):
        """
        This is a workflow graph.
        """
        # Step 1: Break down the problem into detailed steps with reasoning
        analysis1 = await self.custom1(instruction="Can_you_solve_this_problem_by_
            breaking_it_down_into_detailed_steps_and_explaining_the_reasoning_behind_
            each_step?")

        # Step 2: Explain how to solve the problem with clear reasoning for each step (
            independent second analysis)
        analysis2 = await self.custom2(instruction="Explain_how_to_solve_the_problem_
            with_clear_reasoning_for_each_step.")

        # Step 3: Use programmer to write and execute code based on first analysis
        program_solution1 = await self.programmer(analysis=analysis1)

        # Step 4: Use programmer to write and execute code based on second analysis
        program_solution2 = await self.programmer(analysis=analysis2)

        # Step 5: Ensemble the two program solutions to select the best one
        ensembled_solution = await self.sc_ensemble(solutions=[program_solution1,
            program_solution2])

        # Step 6: Review the ensembled solution to regenerate improved solution
        final_solution = await self.review(pre_solution=ensembled_solution)

        return final_solution
```

### ❓ Workflow for Natural Questions (NQ)

```python
class Workflow:
    def __init__(
        self,
        problem
    ) -> None:
        self.problem = problem
        self.custom1 = operator.Custom("qwen/qwen-2.5-72b-instruct", self.problem)
        self.custom2 = operator.Custom("gpt-4o-mini", self.problem)
        self.sc_ensemble = operator.ScEnsemble("gpt-4o", self.problem)
        self.review = operator.Review("qwen/qwq-32b", self.problem)
        self.answer_generate = operator.AnswerGenerate("gpt-4o-mini", self.problem)

    async def run_workflow(self):
        """
```

```
        This is a workflow graph.
        """
        # Generate multiple candidate solutions with different instructions
        candidate1 = await self.custom1("Generate a detailed answer with reasoning.")
        candidate2 = await self.custom2("Provide a concise direct answer.")

        # Ensemble to select the best candidate
        best_solution = await self.sc_ensemble([candidate1, candidate2])

        # Review the best solution to improve it
        reviewed_solution = await self.review(best_solution)

        # Generate final answer based on the reviewed solution
        final_answer = await self.answer_generate()

        return final_answer
```

# D  CASE STUDIES OF WORKFLOW RECTIFICATION

We present two case studies demonstrating the Rectifier agent's capability to diagnose and repair failures in real-time.

## D.1  CASE 1: HANDLING A FAILED ENSEMBLE SELECTION IN A CODING TASK(MBPP)

❗ Original Workflow Code (Before Rectification)

Listing 1: Original workflow that crashes when ensemble selection fails.

```
# [Original Python code from the "generated_workflow" log.]
# Flaw: This workflow implicitly assumes that 'sc_ensemble' will always return a valid,
      non-empty solution.
# If the selection process fails and returns an empty string, the subsequent 'KeyError'
      crashes the system.
class Workflow:
    def __init__(
        self,
        problem
    ) -> None:
        self.problem = problem
        self.code_generate_1 = operator.CustomCodeGenerate("llm_symbol", self.problem)
        self.code_generate_2 = operator.CustomCodeGenerate("llm_symbol", self.problem)
        self.code_generate_3 = operator.CustomCodeGenerate("llm_symbol", self.problem)
        self.sc_ensemble = operator.ScEnsemble("llm_symbol", self.problem)
        self.test = operator.Test("llm_symbol", self.problem)

    async def run_workflow(self):
        solution_list = []
        for _ in range(3):
            solution = await self.code_generate_1(instruction="Please analyze the
                problem carefully and generate the code solution step by step.")
            solution_list.append(solution)

        ensembled_solution = await self.sc_ensemble(solutions=solution_list)
        tested_solution = await self.test(solution=ensembled_solution)

        return tested_solution
```

🐞 Error Message

Listing 2: Error log showing a KeyError during ensemble execution.

```
# [Relevant traceback from the "error" log.]
# Analysis: The traceback indicates a 'KeyError: ''. This occurs within the '
      sc_ensemble' operator
# when the underlying selection mechanism fails to choose a candidate and returns an
      empty string,
# which is then used as a dictionary key, causing the crash.
```

```
Traceback (most recent call last):
  File "workflow_executor.py", line 138, in execute_from_text
    result = loop.run_until_complete(instance())
  ...
  File "tmp_workflow.py", line 26, in run_workflow
    ensembled_solution = await self.sc_ensemble(solutions=solution_list)
  File "operator.py", line 231, in __call__
    return solutions[answer_mapping[answer]]
KeyError: ''
```

### ✔ Revised Workflow Code (After Rectification)

Listing 3: Workflow code after being corrected by the Rectifier agent.

```python
# [Revised Python code with added robustness.]
# The Rectifier agent identified that the workflow lacked a fallback mechanism.
# It introduced a validation step to ensure the program continues even if the ensemble
    operator fails.
class Workflow:
    def __init__(
        self,
        problem
    ) -> None:
        self.problem = problem
        self.code_generate_1 = operator.CustomCodeGenerate("llm_symbol", self.problem)
        self.code_generate_2 = operator.CustomCodeGenerate("llm_symbol", self.problem)
        self.code_generate_3 = operator.CustomCodeGenerate("llm_symbol", self.problem)
        self.sc_ensemble = operator.ScEnsemble("llm_symbol", self.problem)
        self.test = operator.Test("llm_symbol", self.problem)

    async def run_workflow(self):
        solution_list = []
        for _ in range(3):
            solution = await self.code_generate_1(instruction="Please␣analyze␣the␣
                problem␣carefully␣and␣generate␣the␣code␣solution␣step␣by␣step.")
            solution_list.append(solution)

        ensembled_solution = await self.sc_ensemble(solutions=solution_list)

        # --- RECTIFIER'S FIX START ---
        # Add a validation step to handle potential ensemble failure.
        # If the ensembled solution is invalid (e.g., None or empty),
        # it robustly falls back to the first generated candidate.
        if not ensembled_solution or not ensembled_solution.strip():
            ensembled_solution = solution_list[0]
        # --- RECTIFIER'S FIX END ---

        tested_solution = await self.test(solution=ensembled_solution)

        return tested_solution
```

## D.2 CASE 2: RECTIFYING AN OPERATOR WITH MALFORMED STRUCTURED OUTPUT

### ❗ Original Workflow Code (Before Rectification)

Listing 4: Original workflow expecting structured data from a Review operator.

```python
# [Original Python code from the "generated_workflow" log.]
# Flaw: This workflow assumes that the 'review1' and 'review2' operators will always
    return a valid structured object with specific keys. It lacks a mechanism to handle
    cases where an operator
# returns malformed or empty data, leading to a hard crash during data validation.
class Workflow:
    def __init__(
        self,
        problem
    ) -> None:
        self.problem = problem
        self.gid = None
        self.custom1 = operator.Custom("llm_symbol", self.gid, self.problem)
        self.custom2 = operator.Custom("llm_symbol", self.gid, self.problem)
```

```
        self.programmer = operator.Programmer("llm_symbol", self.gid, self.problem)
        self.review1 = operator.Review("llm_symbol", self.gid, self.problem)
        self.review2 = operator.Review("llm_symbol", self.gid, self.problem)
        self.sc_ensemble = operator.ScEnsemble("llm_symbol", self.gid, self.problem)

    async def run_workflow(self):
        analysis = await self.custom1(instruction="...")
        refined_analysis = await self.custom2(instruction="...")
        program_solution1 = await self.programmer(analysis=analysis)

        # This line is expected to return a structured object but fails to do so.
        reviewed_solution1 = await self.review1(pre_solution=program_solution1)

        program_solution2 = await self.programmer(analysis=refined_analysis)
        reviewed_solution2 = await self.review2(pre_solution=program_solution2)

        final_solution = await self.sc_ensemble(solutions=[reviewed_solution1,
            reviewed_solution2])
        return final_solution
```

### 🐛 Error Message

Listing 5: Error log showing a Pydantic validation error.

```
# [Relevant traceback from the "error" log.]
# Analysis: The error is a `pydantic.ValidationError`. It clearly states that the data
    received
# from the `Review` operator is missing mandatory fields: 'revised_solution' and '
    thought'.
# This indicates that the LLM powering the operator failed to generate its output in
    the
# required structured format.

1 validation error for ReviewOp_AN
  Value error, Missing fields: {'revised_solution', 'thought'} [type=value_error,
      input_value={}, input_type=dict]
    For further information visit https://errors.pydantic.dev/2.11/v/value_error
```

### ✓ Revised Workflow Code (After Rectification)

Listing 6: Workflow code after the Rectifier agent bypassed the faulty operator.

```
# [Revised Python code demonstrating a smart rectification strategy.]
# The Rectifier agent diagnosed that the `Review` operators were unreliable.
# Instead of trying to fix the operator itself, it adapted the workflow's logic to
    bypass
# the faulty review steps, ensuring the overall process can complete successfully.
class Workflow:
    def __init__(
        self,
        problem
    ) -> None:
        self.problem = problem
        self.gid = None
        self.custom1 = operator.Custom("llm_symbol", self.gid, self.problem)
        self.custom2 = operator.Custom("llm_symbol", self.gid, self.problem)
        self.programmer = operator.Programmer("llm_symbol", self.gid, self.problem)
        self.review1 = operator.Review("llm_symbol", self.gid, self.problem)
        self.review2 = operator.Review("llm_symbol", self.gid, self.problem)
        self.sc_ensemble = operator.ScEnsemble("llm_symbol", self.gid, self.problem)

    async def run_workflow(self):
        analysis = await self.custom1(instruction="...")
        refined_analysis = await self.custom2(instruction="...")
        program_solution1 = await self.programmer(analysis=analysis)
        program_solution2 = await self.programmer(analysis=refined_analysis)

        # --- RECTIFIER'S FIX START ---
        # The Rectifier identified that the `review` steps were failing due to format
            validation errors.
        # As a recovery strategy, it has modified the workflow to bypass the unreliable
            review operators
```

```
        # and directly pass the initial programmatic solutions to the ensemble stage.
        #
        # reviewed_solution1 = await self.review1(pre_solution=program_solution1)  <-
              Bypassed
        # reviewed_solution2 = await self.review2(pre_solution=program_solution2)  <-
              Bypassed

        final_solution = await self.sc_ensemble(solutions=[program_solution1,
            program_solution2])
        # --- RECTIFIER'S FIX END ---

        return final_solution
```

## D.3 CASE 3: DYNAMIC EXTENSION OF SEARCH BUDGET IN INFORMATION RETRIEVAL

In complex information retrieval tasks, a rigid iteration limit often prematurely terminates the search process before the target information is located. In this case, the Rectifier detects that the system failed to retrieve the answer due to an exhausted search budget and dynamically extends the `max_iterations` parameter to allow for deeper exploration.

❗ Original Workflow Code (Before Rectification)

Listing 7: Original workflow with a restrictive iteration limit.

```python
# [Original Python code from the "generated_workflow" log.]
# Flaw: The 'max_iterations' is set to a conservative value of 3.
# For obscure queries, 3 rounds are insufficient to traverse the
# necessary link depth, leading to a "Not Found" result.
class Workflow:
    def __init__(self, problem) -> None:
        self.problem = problem
        self.custom = operator.Custom("gpt-4o", self.problem)
        self.search = operator.Search("qwen/qwq-32b", self.problem)
        self.browser = operator.Browser("gpt-4o-mini", self.problem)
        self.answer_generate = operator.AnswerGenerate("qwen/qwen-2.5-72b-instruct",
            self.problem)

    async def run_workflow(self):
        search_history = set()
        collected_evidence = ""

        # FAILURE POINT: 3 iterations are too few for this complex query
        max_iterations = 3

        iteration = 0
        query = self.problem

        while iteration < max_iterations:
            iteration += 1
            # Reflection: plan and rewrite query to optimize search
            reflection_prompt = f"<think>Iteration {iteration}: Plan...</think>"
            rewritten_query = await self.custom(instruction=reflection_prompt)

            # ... [Code logic for search and browse omitted for brevity] ...

            search_results = await self.search(query=rewritten_query, top_k=5)
            if not search_results: break

            # Extract docids and browse for full content
            new_evidence = []
            for result in search_results:
                # ... [Browsing logic] ...
                pass

            # Accumulate evidence
            if new_evidence:
                collected_evidence += "\n\n".join(new_evidence) + "\n\n"

            # Reflection: check if sufficient evidence collected
            decision = await self.custom(instruction="...is it sufficient?...")
            if "yes" in decision.lower(): break

        # Final answer generation
```

```
    if collected_evidence.strip():
        solution = await self.answer_generate(context=collected_evidence.strip())
    else:
        solution = "Information_Not_Found_in_Context"

    return solution
```

### 🐛 Runtime Log

Listing 8: Runtime log showing "Information Not Found" due to timeout.

```
# [Relevant log from the execution environment.]
[System] Max iterations (3) reached. Terminating loop.
[Result] "Information_Not_Found_in_Context"

# [Rectifier Diagnosis]
# Task failed due to insufficient evidence. The agent was making progress
# but ran out of steps. Action: Extend search budget.
```

### ✔ Revised Workflow Code (After Rectification)

Listing 9: Workflow code with extended iteration limit.

```python
# [Revised Python code.]
# The Rectifier agent identified that the evidence was insufficient and the
# loop terminated due to the iteration count. It increased 'max_iterations'
# from 3 to 6 to provide a wider temporal window for exploration.
class Workflow:
    def __init__(self, problem) -> None:
        self.problem = problem
        self.custom = operator.Custom("gpt-4o", self.problem)
        self.search = operator.Search("qwen/qwq-32b", self.problem)
        self.browser = operator.Browser("gpt-4o-mini", self.problem)
        self.answer_generate = operator.AnswerGenerate("qwen/qwen-2.5-72b-instruct",
            self.problem)

    async def run_workflow(self):
        search_history = set()
        collected_evidence = ""

        # --- RECTIFIER'S FIX START: PARAMETER TUNING ---
        # Increased max_iterations from 3 to 6 to allow deeper search.
        max_iterations = 6
        # --- RECTIFIER'S FIX END ---

        iteration = 0
        query = self.problem

        while iteration < max_iterations:
            iteration += 1
            # ... [Standard search and browse loop continues] ...
            # ...
            pass

        # Final answer generation
        if collected_evidence.strip():
            solution = await self.answer_generate(context=collected_evidence.strip())
        else:
            solution = "Information_Not_Found_in_Context"

        return solution
```

## D.4 CASE 4: UPGRADING MODEL CAPACITY TO HANDLE CONTEXT OVERLOAD

In deep research tasks involving extensive information retrieval, lightweight models often fail to generate a valid response due to token limits or an inability to filter noisy web content. In this case, the Rectifier diagnoses a silent failure ("Empty Response") and applies a dual-fix strategy: upgrading the LLM backbone and injecting strict prompt constraints to improve robustness.

---

⚠ Original Workflow Code (Before Rectification)

Listing 10: Original workflow using a lightweight model for final answer generation.

```python
# [Original Python code from the "generated_workflow" log.]
# Flaw: The workflow assigns 'gpt-4o-mini' to the 'AnswerGenerate' operator.
# In BrowseComp+, the accumulated evidence often contains noisy HTML artifacts
# that overwhelm smaller models, causing them to return empty responses.
class Workflow:
    def __init__(self, problem) -> None:
        self.problem = problem
        self.custom = operator.Custom("gpt-4o", self.problem)
        self.search = operator.Search("gpt-4o-mini", self.problem)
        self.browser = operator.Browser("qwen/qwen3-14b", self.problem)
        # FAILURE POINT: Weak model assigned to final generation
        self.answer_generate = operator.AnswerGenerate("gpt-4o-mini", self.problem)

    async def run_workflow(self):
        # ... [Initialization omitted] ...
        collected_evidence = ""

        # ... [Search and browsing loop omitted] ...

        # Final answer generation
        if collected_evidence.strip():
            # This call fails silently due to context overload
            solution = await self.answer_generate(context=collected_evidence.strip())
        else:
            solution = "Information_Not_Found_in_Context"

        return solution
```

---

🐛 Runtime Log

Listing 11: Runtime log capturing the silent failure of the LLM API.

```
# [Relevant log from the execution environment.]
[AnswerGenerate] Using model: gpt-4o-mini with max_tokens: 4096
[AnswerGenerate] WARNING: LLM returned empty response!
...
[SystemError] RuntimeError: Answer generation failed. Received empty response.
```

---

✅ Revised Workflow Code (After Rectification)

Listing 12: Workflow code after Rectifier intervention (Model Upgrade + Prompt Fix).

```python
# [Revised Python code with Rectifier intervention.]
# The Rectifier agent identified the failure as a capacity issue. It applied two fixes:
# 1. Upgraded the backbone to 'gpt-4o'.
# 2. Modified the input context with a strict prompt wrapper to handle noise.
class Workflow:
    def __init__(self, problem) -> None:
        self.problem = problem
        self.custom = operator.Custom("gpt-4o", self.problem)
        self.search = operator.Search("gpt-4o-mini", self.problem)
        self.browser = operator.Browser("qwen/qwen3-14b", self.problem)

        # --- RECTIFIER'S FIX 1: MODEL UPGRADE ---
        # Switched from 'gpt-4o-mini' to 'gpt-4o' for better long-context handling.
        self.answer_generate = operator.AnswerGenerate("gpt-4o", self.problem)
        # --- RECTIFIER'S FIX END ---

    async def run_workflow(self):
        # ... [Search loop omitted] ...

        # Final answer generation
        if collected_evidence.strip():
            # --- RECTIFIER'S FIX 2: PROMPT REFINEMENT ---
            # The Rectifier injected a wrapper to help the model focus.
            refined_context = (
                "IMPORTANT_INSTRUCTION:_Ignore_all_HTML_tags_and_irrelevant_headers._"
```

```
                    "Focus_only_on_factual_claims.\n\n"
                    f"{collected_evidence.strip()}"
            )
            solution = await self.answer_generate(context=refined_context)
        else:
            solution = "Information_Not_Found_in_Context"

        return solution
```

