# OpenReview forum: "MAS$^2$: Self-Generative, Self-Configuring, Self-Rectifying Multi-Agent Systems"
_ICLR.cc/2026/Conference — ICLR 2026 Poster_

### Official Review · Reviewer_3oGm · 2025-10-30

**Soundness:** 3
**Presentation:** 2
**Contribution:** 3
**Rating:** 6
**Confidence:** 4

**Summary:**

This paper proposes MAS², a "recursive self-generative" multi-agent paradigm built on a tri-agent architecture: a generator designs high-level workflow templates based on tasks, an implementer assigns specific LLM backbones to turn templates into executable systems, and a rectifier monitors real-time operations and dynamically rectifies configurations. The paper also introduces the Collaborative Tree Optimization (CTO) framework, which specializes meta-agents by collecting trajectory data via a collaborative decision tree, generating preference signals through path credit propagation, and optimizing policies via value-guided preference alignment. Experiments show the effectiveness of the proposed model.

**Strengths:**

1. The Collaborative Tree Optimization (CTO) framework addresses meta-agent training challenges (e.g., ambiguous rewards, low data efficiency) via collaborative decision trees, path credit propagation, and value-guided alignment. It attributes terminal feedback to upstream decisions and prioritizes high-impact preferences, enabling efficient specialization of each meta-agent.

2. Experiments cover 8 benchmarks across 4 domains (multi-hop search, deep research, code generation, mathematical reasoning) and 13 baselines.

3. The overall framework seems reasonable and effective.

**Weaknesses:**

1. Rectifier Threshold Lacks Clear Justification: The rectifier’s trigger relies on a resource budget threshold θ_C (for excessive consumption), but the paper does not explain how θ_C is determined, whether it is set manually or adaptively. No sensitivity analysis is provided to show how θ_C adjustments affect system performance, leaving uncertainty about its practical flexibility.

2. CTO Parameter Selection Lacks Optimization Proof: The CTO framework uses fixed parameters (e.g., 4 candidate templates for the generator, 2 instantiations for the implementer) in experiments. The paper fails to justify why these values are optimal or how changing them (e.g., more templates) might impact training efficiency or trajectory quality.

3. Dynamic Interference Scenarios Are Undertested: While MAS² claims to handle real-world dynamism (e.g., tool crashes), experiments only use static benchmarks. No tests simulate dynamic disruptions (e.g., random tool failures during execution), making it hard to fully verify its robustness in actual unstable environments.

4. The paper lacks key engineering insights for real-world use, such as the computational resources (GPU type, training time) needed for meta-agent training or the latency of generating executable MAS. These omissions hinder practical deployment planning for industrial users.

**Questions:**

1. For the rectifier’s threshold θ_C, could you clarify whether it is set manually or adaptively, and provide sensitivity analysis on how θ_C adjustments affect system performance?

2. Regarding the CTO framework’s fixed parameters (4 generator templates, 2 implementer instantiations), could you explain the rationale for these values and test results of other parameter combinations?

3. Since experiments lack dynamic interference simulations, could you add tests (e.g., random tool failures) to verify MAS²’s robustness in unstable environments?

4. For engineering deployment, could you provide details like meta-agent training resources  and executable MAS generation latency?

---

> ### Author Response · Authors · 2025-11-22
> **[Part 1/2] Author Response**
>
> Thank you immensely for your time and efforts, as well as the helpful and constructive feedback! Here, we give point-by-point responses to your comments.
>
>
>
> ---
> > **`Weakness 1: Rectifier Threshold Lacks Clear Justification`** The rectifier’s trigger relies on a resource budget threshold θ_C, but the paper does not explain how θ_C is determined, whether it is set manually or adaptively. No sensitivity analysis is provided to show how θ_C adjustments affect system performance.
>
> We sincerely apologize for any confusion. Recall that the rectifier agent is activated under two conditions:
>
> * **System failure**, such as search-engine errors, code-execution failures, etc.
> * **Cumulative resource cost $C(s_t)$ exceeding the budget $\theta_C$**.
>   Specifically, we measure $C(s_t)$ and $\theta_C$ using the **number of execution steps**. If a generator’s multi-agent system has executed $s'$ steps without producing a final result (which is treated as exceeding $\theta_C$), the rectifier intervenes and executes Eq. (6). For *multi-hop search* and *deep research* datasets, we set $s' = 6$; for *code* and *mathematical reasoning* datasets, we set $s' = 4$, since these tasks are generally simpler.
>
> To quantify the effect of the budget $\theta_C$ on performance, we provide the following ablation:
>
> |$\theta_C$|2|4|6|8|10|
> |-|-|-|-|-|-|
> |NQ|77.2|79.8|79.1|78.1|76.4|
> |BrowseComp+|17.0|18.5|19.7|18.5|17.4|
> |MBPP|83.6|85.1|83.6|81.7|81.7|
> |MATH|70.1|71.3|67.3|64.7|64.7|
>
> We observe that when $\theta_C$ is set too small (triggering overly frequent rectifier interventions) or too large (triggering overly infrequent ones), performance degrades. The former disrupts multi-agent execution, while the latter fails to correct errors promptly. Our chosen values $s' \in \{4,6\}$ lie near the performance peak and require no extensive tuning. These clarifications have been added on Line 222 of the revised manuscript.
>
> Finally, we note that the resource function $C(\cdot)$ used in Eq. (5) differs from the $C(\cdot)$ in Eq. (8): the former counts execution steps, whereas the latter measures the system-level LLM API cost (as stated on Line 253). We have explicitly clarified this distinction in the updated manuscript.
>
>
>
> ---
> > **`Weakness 2: CTO Parameter Selection Lacks Optimization Proof`** The CTO framework uses fixed parameters (e.g., 4 candidate templates for the generator, 2 instantiations for the implementer) in experiments. The paper fails to justify why these values are optimal or how changing them (e.g., more templates) might impact training efficiency or trajectory quality.
>
> To address your concern, we aim to (1) clarify quantitatively how the number of templates $K$ and the number of instantiations $N$ influence the CTO algorithm, and (2) provide experimental results for different choices of $K$ and $N$.
>
> **What do $K$ and $N$ affect?**
>
> Recall that for each question, the generator produces $K$ multi-agent templates. The implementor then expands these into $K \cdot N$ instantiations, and the rectifier supervises and corrects the executions of all $K \cdot N$ multi-agent systems in real time.
>
> After computing node values via Eq. (9), the generator can obtain up to $\binom{K}{2}$ pairs of preference data, and the implementer can obtain up to $K\cdot\binom{N}{2}$ preference pairs. Thus, the primary effect of increasing $K$ or $N$ is an increase in the amount of data collected for CTO, which indirectly effects the performance of the trained meta agents.
>
>
> **How do different $K$ and $N$ influence data volume and performance?**
>
> To quantify this relationship, we additionally evaluate two settings $K{=}2, N{=}2$ and $K{=}2, N{=}4$ and report the full results below:
>
> *[Parameter analysis of CTO. We use 300 samples for data collection.]*
> |Setting|Total traj. rollouts|#Generator data|#Implementor data|#Rectifier Data|Data collection wall-clock time|HotpotQA (Acc.)|Bamboogle (Acc.)|NQ (Acc.)|
> |-|-|-|-|-|-|-|-|-|
> |$K=2,N=2$|1200|606|1009|1130|8.3h|79.3|57.0|73.7|
> |$K=2,N=4$|2400|609|10235|2267|17.2h|85.6|65.6|75.0|
> |$K=4,N=2$ (original setting)|2400|1583|2371|2189|14.8h|87.0|64.8|77.0|
>
> As expected, enlarging either $K$ or $N$ increases the volume of collected preference data, leading to consistent performance gains. In general, increasing $K$ mainly expands generator data, while increasing $N$ more strongly amplifies implementor data.
>
> Between the two configurations with equal rollout budgets ($K{=}4,N{=}2$ vs. $K{=}2,N{=}4$), the former performs slightly better due to a more balanced allocation of training data across all three meta-agents. Nonetheless, both larger configurations outperform $K{=}2,N{=}2$, and we expect performance to continue improving with further scaling.

---

> ### Author Response · Authors · 2025-11-22
> **[Part 2/2] Author Response**
>
> > **`Weakness 3: Dynamic Interference Scenarios Are Undertested`**: While MAS² claims to handle real-world dynamism (e.g., tool crashes), experiments only use static benchmarks. No tests simulate dynamic disruptions (e.g., random tool failures during execution), making it hard to fully verify its robustness in actual unstable environments.
>
>
> Thank you for this valuable question. To address your concern, we respond from two perspectives:
>
> * **MAS²’s benchmarks naturally and frequently trigger real-world tool errors, and we already provide such cases in the appendix.** Specifically, HotpotQA, Bamboogle, NQ, and BrowseComp+ involve extensive calls to search engines and retrieval tools, where tool crashes are common. Likewise, HumanEval and MBPP frequently invoke code interpreters and often encounter execution errors. In Appendix D, we have already shown concrete examples where the rectifier successfully handles ``Pydantic validation errors`` and `Python KeyError` exceptions. Therefore, we respectfully argue that MAS² has already demonstrated its capacity to correct the types of real-world dynamism (e.g., tool crashes) you mentioned.
> * **We have further added more complex tool-failure case studies.** To more fully address your concern, we additionally include two rectifier case studies from the more challenging deep-research benchmark BrowseComp+, showing how the rectifier repairs complex search-tool crashes (context overload, search tool time error). These examples are now incorporated into the revised manuscript as Appendix D.3 and Appendix D.4.
>
> We sincerely hope this resolves your question.
>
>
>
>
> ---
> > **`Weakness 4`**: The paper lacks key engineering insights for real-world use, such as the computational resources (GPU type, training time) needed for meta-agent training or the latency of generating executable MAS.
>
>
> Thank you for your question! Following your valuable suggestion, we now report the key computational metrics of MAS² as follows:
>
> *[Training cost of MAS² on QA domain.]*
> | Metric |($K=4,N=2$ setting)|
> | - | - |
> | Data collection wall-clock time| 14.8h |
> | Total GPU hours | 4.8h |
> | GPU hours (generator)|0.9h |
> | GPU hours (implementor)| 3.8h|
> | GPU hours (rectifier) | 2.1h |
>
> As explicitly stated on Line 322, all three meta-agents, are trained using LoRA (rank = 8) on 4×L40(48G) GPUs. Thus, the overall computational requirements remain modest.
>
> We also report inference-related metrics as below:
>
> *[Inference cost comparison of MAS² and ScoreFlow.]*
> |Dataset| Performance (%)|MAS generation latency per query (total) | latentcy (generator)| latentcy (implementor)| Execution Latency per query|
> |-|-|-|-|-|-|
> |HotpotQA (MAS²)|89.3|8.67s|5.90s|2.77s| 31.2s|
> |Bamboogle (MAS²) |67.2|10.80s|7.12s|3.68s|35.8s|
> |HotpotQA (ScoreFlow)| 86.0 | 6.32s|-|-|27.8s|
> |Bamboogle (ScoreFlow)| 64.8| 8.81s|-|-|41.56s|
>
> As shown, the MAS-generation overhead introduced by MAS² is modest relative to the full MAS execution cost. When compared to ScoreFlow, MAS² increases MAS-generation latency by only about 2 seconds while achieving substantially higher accuracy. We respectfully believe this trade-off is both reasonable and desirable.
>
> In accordance with your request, we have updated all of these key details to **Appendix F.1**. Thank you once again for your critical suggestions!
>
>
>
> ---
> > **`Question 1`**: For the rectifier’s threshold θ_C, could you clarify whether it is set manually or adaptively, and provide sensitivity analysis on how θ_C adjustments affect system performance?
>
> Thank you for this insightful question! We have added detailed clarification and a full sensitivity analysis in our response to **`Weakness 1`**.
>
>
> ---
> > **`Question 2`**: Regarding the CTO framework’s fixed parameters (4 generator templates, 2 implementer instantiations), could you explain the rationale for these values and test results of other parameter combinations?
>
> Thank you for raising this important point. Our primary rationale is that with $K = 4$ and $N = 2$, the generator can obtain up to $\binom{K}{2} = 6$ preference pairs per query, and the implementor can obtain up to $K\cdot\binom{N}{2} = 4$ pairs: quantities we consider sufficiently informative for training. We further added the full results for alternative parameter combinations (e.g., $K=2, N=2$ and $K=2, N=4$) in our response to **`Weakness 2`**.
>
>
> ---
> > **`Question 3`**: Since experiments lack dynamic interference simulations, could you add tests (e.g., random tool failures) to verify MAS²’s robustness in unstable environments?
>
> We respectfully refer the reviewer to our response to **`Weakness 3`**.
>
> ---
> > **`Question 4`**: For engineering deployment, could you provide details like meta-agent training resources and executable MAS generation latency?
>
> We have provided a comprehensive breakdown of training and inference resources in our response to **`Weakness 4`**. Thanks immensely for your thoughtful questions that have improved the clarity and completeness of our manuscript!

---

> ### Author Response · Authors · 2025-11-25
> **Thank you & Looking forward to further discussion**
>
> Dear Reviewer **3oGm**,
>
> We deeply appreciate your dedication to engaging in author–reviewer discussions. Recognizing the existing discrepancies in the understanding of our manuscript, we have outlined your key concerns and our responses for enhanced communication:
> 1. **Regarding rectifier threshold justification `Weakness 1` `Question 1`**: We clarified how $θ_C$ is defined and provided full sensitivity analyses.
> 2. **Regarding CTO parameter selection `Weakness 2` `Question 2`**: We comprehensively analyzed how template count $K$ and instantiation count $N$ influence data and performance.
> 3. **Regarding robustness under dynamic interference `Weakness 3` `Question 3`**: We highlighted real tool-failure cases already present in our benchmarks and added new BrowseComp+ case studies demonstrating MAS²’s robustness to tool crashes.
> 4. **Regarding computational details `Weakness 4` `Question 4`**: We reported full training and inference metrics, including GPU hours, training resources, and MAS-generation latency.
>
>
> For other issues not mentioned here, please refer to our detailed rebuttal response. We sincerely hope this addresses your concerns! We humbly look forward to further discussion with you.
>
> Warm regards,
> **Authors**

---

### Official Review · Reviewer_UFEM · 2025-11-01

**Soundness:** 3
**Presentation:** 3
**Contribution:** 2
**Rating:** 4
**Confidence:** 4

**Summary:**

This paper introduces MAS2, a new paradigm for automatic multi-agent system (MAS) construction powered by large language models (LLMs). Unlike prior methods that generate a system once and execute it statically, MAS2 enables a meta MAS to recursively generate, configure, and rectify MASs.

**Strengths:**

1. Novel Recursive MAS Paradigm: The paper introduces MAS2, a self-generative, self-configuring, self-rectifying multi-agent paradigm, moving beyond the dominant “generate once and deploy” paradigm. This conceptual shift is original and timely for dynamic real-world settings.
2. Agent Meta Architecture: The generator–implementer–rectifier division of labor provides clear modularity: generator → architectural workflow; implementer → LLM backbone assignment; rectifier → online adaptation
This creates specialization, interpretability, and controllability.

**Weaknesses:**

1. In the Introduction, the authors claim that previous methods based on Bayesian optimization, GNNs, and MCTS suffer from the limitations of predefined search spaces and atomic operators. However, in Figure 2, the workflow template of MAS² also appears to be composed of similar atomic operators such as "manager", "coder," and "summarizer." This design seems structurally indistinguishable from earlier works, weakening the claimed motivation and novelty.
2. The paper lacks crucial implementation details: How large is the dataset used to train the generator, implementer, and rectifier agents? Were any of the training tasks reused during evaluation, or are they strictly separated?
3. On the BrowseComp+ benchmark, MAS² shows only a 3.6% improvement over the CoT baseline, does it suggests limited effectiveness when handling high-difficulty, complex reasoning tasks.
4. Inconsistent Benchmark Reporting: The number of benchmarks is contradictory — abstract (7), §4.1 (8), and contribution summary (6), while Table 1 actually contains seven columns. This inconsistency undermines clarity of experimental coverage.

**Questions:**

CTO calibration: How are Rₚ(τ) (success condition) and θ₍C₎ (cost threshold) determined? Are they fixed, learned, or task-adaptive? What is the effect of varying them?

---

> ### Author Response · Authors · 2025-11-22
> **[Part 1/2] Author Response**
>
> We sincerely thank you for the thoughtful and constructive reviews of our manuscript! Based on your questions and recommendations, we give point-by-point responses to your comments and describe the revisions we made to address them.
>
>
>
> ---
> > **`Weakness 1`**: In the Introduction, the authors claim that previous methods based on Bayesian optimization, GNNs, and MCTS suffer from the limitations of predefined search spaces and atomic operators. However, in Figure 2, the workflow template of MAS² also appears to be composed of similar atomic operators such as "manager", "coder," and "summarizer." This design seems structurally indistinguishable from earlier works, weakening the claimed motivation and novelty.
>
>
> Thank you for this exceptionally insightful question! We hope our response can more precisely articulate the distinction between our method and previous Bayesian/GNN/MCTS-based approaches.
>
> To begin, recall that classical multi-agent optimization frameworks (e.g., AgentSquare, G-Designer, and AFlow) typically rely on a fixed library of agentic operators (e.g., CoT, self-refine, multi-agent debate). Their optimization procedure amounts to searching over different permutations or combinations of these predefined operators, and thus their search space is inherently limited.
>
> Subsequent agent-based multi-agent optimization methods, such as MAS-GPT and FlowReasoner, introduced a different paradigm: “given a problem, use an LLM agent to synthesize an entire multi-agent system.” One key advantage of this paradigm is that **the generated system is no longer constrained by a predefined operator set**. Instead, the agent can freely invent new structures tailored to the task at hand. For example, FlowReasoner-14B creates a novel GRAPH_SOLUTION_EXPLANATION operation when handling an HTML parsing task, an operator not present in any predefined library.
>
> **MAS² preserves exactly this advantage.** The multi-agent systems it generates are not mere compositions of existing operators; rather, the generator agent can dynamically create new operators, write new prompts, and assemble new architectures fully customized to the scenario. **Do we have concrete evidence for this? Yes!** As shown in Line 1020, the MAS²-generated system for BrowseComp+ introduces a new browse-reflection operation specifically designed for iterative search and browsing (it is quite different from the vanilla self-reflection operator), rather than reusing any existing operator template, which would surely fail to generalize across all settings.
>
> Once again, we are sincerely grateful for your thoughtful question. We have updated the caption of Figure 2 in the revised manuscript to clarify this distinction.
>
>
>
> ---
> > `Weakness 2`: The paper lacks crucial implementation details: How large is the dataset used to train the generator, implementer, and rectifier agents? Were any of the training tasks reused during evaluation, or are they strictly separated?
>
>
> To address your concern, we provide a detailed breakdown of the training and test splits for each dataset:
>
>
> |Dataset|HotpotQA|Bamboogle|NQ|BrowseComp+|MBPP|MATH|HumanEval|
> |-|-|-|-|-|-|-|-|
> |#Training samples|1000|-|1100|664|700|484|-|
> |#Test samples|300|125|300|166|273|121|164|
>
> As described in Section 3.2, we apply the data-curation pipeline to the *training* portion of each dataset to collect preference trajectories and train MAS², after which we evaluate solely on the *test* portion. We respectfully emphasize that the training and evaluation splits are strictly separated, with **no overlap** of any kind. We also supplment these details in Appendix E.2 of our updated manuscript; thank you again for your thoughtful comment!

---

> ### Author Response · Authors · 2025-11-22
> **[Part 2/2] Author Response**
>
> > **`Weakness 3`**: On the BrowseComp+ benchmark, MAS² shows only a 3.6% improvement over the CoT baseline, does it suggests limited effectiveness when handling high-difficulty, complex reasoning tasks.
>
> We would like to address your concern from two perspectives:
>
> - **MAS² already delivers substantial gains over existing multi-agent system baselines.**  As shown in Table 1, prior multi-agent systems perform extremely poorly on the challenging BrowseComp+ benchmark: ADAS even underperforms the Avg. baseline by 2.5%, and ScoreFlow improves by only 0.9%. In contrast, MAS² improves over the Avg. baseline by 10.2%, demonstrating clear and consistent generalization across task domains.
> - **The modest improvement over the CoT baseline is primarily due to the benchmark’s inherent difficulty and its dependence on stronger LLM backbones.**  We respectfully believe that BrowseComp+ is intrinsically difficult and requires a significantly stronger backbone to exhibit large performance gains. For instance, in the original BrowseComp+ paper, Qwen3-32B achieves only 9.16% accuracy, and SearchR1-32B reaches 10.36%. In comparison, GPT-5 reaches 70.12% and OpenAI o3 reaches 63.49%. This suggests that the modest improvement of MAS² stems from limitations of the underlying model, rather than incapability of the framework itself. With stronger LLMs, MAS² can leverage their capabilities and produce **substantially greater improvements**.
>
>  To validate the above claim, we include BrowseComp+ results in Table 2 of our model-generalization study, where we do *not* retrain MAS² but simply add Qwen3-Coder, GPT-5-Mini, and Gemini-2.5-Pro to the implementor’s model pool at inference time:
>
>   | BrowseComp+ | Acc. (%) |
>   |-------------|----------|
>   | [A] Qwen3-Coder | 27.6 |
>   | [B] GPT-5-Mini | 34.6 |
>   | [C] Gemini-2.5-Pro | 30.2 |
>   | MAS² (+A/B/C) | 40.2 |
>
>   As we can see, MAS² equipped with stronger LLMs surpasses each individual model, achieving gains of +10.0% over Gemini-2.5-Pro and +12.6% over Qwen3-Coder.  We sincerely hope this can well respond to the performance gain concerns of our method.
>
>
> ---
> > **`Weakness 4`**: Inconsistent Benchmark Reporting: The number of benchmarks is contradictory — abstract (7), §4.1 (8), and contribution summary (6), while Table 1 actually contains seven columns. This inconsistency undermines clarity of experimental coverage.
>
> Thank you for pointing this out! We have fully standardized this value to seven across the revised manuscript to avoid any potential confusion.
>
>
> ---
> > **`Weakness 5`**: CTO calibration: How are Rₚ(τ) (success condition) and θ₍C₎ (cost threshold) determined? Are they fixed, learned, or task-adaptive? What is the effect of varying them?
>
> Thank you for this thoughtful inquiry and please allow us to respond point-by-point:
>
> - **Success status $R_p(\tau)$ in Eq. (8)** simply indicates whether the task has been successfully solved; for example, whether a mathematical answer is correct or whether a coding solution passes the unit tests. These success criteria follow directly from the definitions provided by each benchmark.
> - **Resource budget $\theta_C$ in Eq (5)**: In Eq. (5), we measure the cumulative cost $C(s_t)$ and its threshold $\theta_C$ using the **number of execution steps**. If a generated multi-agent system executes $s'$ steps without producing a final result (which we treat as exceeding $\theta_C$), the rectifier intervenes and applies Eq. (6). For *multi-hop search* and *deep-research* datasets, we set $s' = 6$; for *code* and *mathematical-reasoning* datasets, we set $s' = 4$, as these tasks are typically simpler. To quantify the effect of the budget $\theta_C$ on performance, we provide the following ablation:
>
> |$\theta_C$|2|4|6|8|10|
> |-|-|-|-|-|-|
> |NQ|77.2|79.8|79.1|78.1|76.4|
> |BrowseComp+|17.0|18.5|19.7|18.5|17.4|
> |MBPP|83.6|85.1|83.6|81.7|1.7|
> |MATH|70.1|71.3|67.3|64.7|64.7|
>
> We observe that when $\theta_C$ is too small (implying excessively frequent rectifier interventions) or too large (implying overly infrequent interventions), performance degrades. We believe the former overly disrupts the multi-agent execution, while the latter fails to correct issues in time. Overall, our chosen settings of $s' \in \\{4, 6\\}$ lie near the peak of the performance curve and require no extensive parameter tuning. We sincerely hope this addresses your concern.

---

> ### Author Response · Authors · 2025-11-25
> **Thank you & Looking forward to further discussion**
>
> Dear Reviewer **UFEM**,
>
> We sincerely appreciate your careful reading and thoughtful feedback on our manuscript! To promote clearer communication, we summarize your main concerns and our corresponding clarifications below:
>
> 1. **Regarding the creativity of MAS² `Weakness 1`**: We clarified that MAS², unlike prior Bayesian/GNN/MCTS frameworks, allows the generator to synthesize entirely new operators rather than relying on a predefined operator set.
> 2. **Regarding implementation details `Weakness 2`**: We supplemented detailed dataset statistics and confirmed that all training and evaluation splits are strictly separated with no overlap.
> 3. **Regarding modest gains on BrowseComp+ `Weakness 3`**: We clarified that BrowseComp+ is inherently difficult under weaker LLMs and showed that MAS² achieves substantially larger gains when equipped with stronger LLMs.
> 5. **Regarding CTO calibration of Rₚ(τ) and θ_C `Weakness 5`**
>    We clarified how these thresholds are determined and added full sensitivity analyses.
>
> For other issues not mentioned here, please refer to our detailed rebuttal response. We sincerely hope this addresses your concerns and warmly welcome further discussion.
>
> Warm regards,
> **Authors**

---

### Official Review · Reviewer_Bhu1 · 2025-11-01

**Soundness:** 3
**Presentation:** 3
**Contribution:** 4
**Rating:** 8
**Confidence:** 3

**Summary:**

This paper introduces MAS2, a novel multi-agent system (MAS) framework endowed with self-generation, self-configuration, and self-correction capabilities for dynamic construction and optimization of multi-agent systems to address complex and evolving task scenarios. MAS2 employs a recursive approach involving a team of agents—generators, executors, and correctors—to build multi-agent systems, thereby overcoming the rigidity of traditional "generate-once-and-deploy" paradigms. Additionally, it leverages collaborative tree optimization (CTO) to enhance the task adaptability and efficiency of meta-agents. MAS2 demonstrates exceptional generality and efficiency across multiple benchmark tasks, particularly in complex domains such as code generation, deep research, and mathematical reasoning.

**Strengths:**

1.The MAS2 paradigm introduces a novel approach to multi-agent systems through recursive self-generation and self-adaptation. The use of a generator-implementer-rectifier tri-agent architecture is a significant advancement over traditional "generate-once-and-deploy" frameworks, enabling dynamic and robust system design.
2.The paper conducts extensive experiments across seven benchmarks, demonstrating MAS2's superior performance in diverse domains such as multi-hop search, deep research, code generation, and mathematical reasoning. The performance improvements (up to 23.8% over SOTA) and cost-efficiency analysis provide strong evidence for the practical utility of MAS2.
3.MAS2 effectively integrates unseen large language models (LLMs) without additional fine-tuning, showcasing its flexibility and scalability. This feature is especially valuable for real-world applications where new models are frequently introduced.

**Weaknesses:**

1. The tri-agent architecture and collaborative tree optimization (CTO) framework, while innovative, introduce significant complexity. The practical challenges of implementing and maintaining such a system, especially in resource-constrained settings, are not thoroughly addressed.
2.Although the rectifier agent is designed to handle runtime errors, the paper provides limited quantitative analysis of how often and effectively the rectifier resolves failures.

**Questions:**

1. While the paper demonstrates MAS2's effectiveness on academic benchmarks, it lacks real-world deployment or interaction scenarios. Is it possible to verify the effectiveness of the method in a real-world dynamic environment, or to provide some examples of real users using multi-agent systems?

---

> ### Author Response · Authors · 2025-11-22
> **[Part 1/2] Author Response**
>
> We sincerely thank you for your careful comments and thorough understanding of our paper! Here we give point-by-point responses to your comments and describe the revisions we made to address them.
>
>
>
> ---
> > **`Weakness 1`**: The tri-agent architecture and collaborative tree optimization (CTO) framework, while innovative, introduce significant complexity. The practical challenges of implementing and maintaining such a system, especially in resource-constrained settings, are not thoroughly addressed.
>
>
> Thank you for raising this important point! We appreciate the opportunity to clarify the practical feasibility of implementing and maintaining the MAS² framework. Concretely, we report the key computational and operational cost of MAS², demonstrating that the system remains lightweight and practical even under constrained resources.
>
> *[Training cost of MAS² on QA domain.]*
> | Metric |($K=4,N=2$ setting)|
> | - | - |
> | Data collection wall-clock time| 14.8h |
> | Total GPU hours | 4.8h |
> | GPU hours (generator)|0.9h |
> | GPU hours (implementor)| 3.8h|
> | GPU hours (rectifier) | 2.1h |
>
> As noted in Line 322, all three meta-agents (generator / implementor / rectifier) are fine-tuned using LoRA (rank = 8) on L40 (48GB) GPUs. This modest configuration suffices for stable convergence, keeping both memory footprint and GPU-hours manageable.
>
>
> We also report inference-related metrics as follows:
>
> |Dataset| Performance (%)|MAS generation latency per query (total) | latentcy (generator)| latentcy (implementor)| Execution Latency per query|
> |-|-|-|-|-|-|
> |HotpotQA (MAS²)|89.3|8.67s|5.90s|2.77s| 31.2s|
> |Bamboogle (MAS²) |67.2|10.80s|7.12s|3.68s|35.8s|
> |HotpotQA (ScoreFlow)| 86.0 | 6.32s|-|-|27.8s|
> |Bamboogle (ScoreFlow)| 64.8| 8.81s|-|-|41.56s|
>
> As shown, the MAS-generation overhead introduced by MAS² is modest relative to the full MAS execution cost. When compared to ScoreFlow, MAS² increases MAS-generation latency by only about 2 seconds while achieving substantially higher accuracy. We respectfully believe this trade-off is both reasonable and desirable.
>
>
> ---
> > **`Weakness 2`**: Although the rectifier agent is designed to handle runtime errors, the paper provides limited quantitative analysis of how often and effectively the rectifier resolves failures.
>
> Thank you for this exceptionally insightful question! We sincerely appreciate your careful reading, which helped us strengthen an important missing component of the manuscript. To address your concern, recall that the rectifier agent is activated under two conditions: (1) when the MAS encounters an explicit system failure such as a tool error, and (2) when MAS execution exceeds the resource budget $\theta_C$, measured concretely by a threshold on execution steps. To quantify its contribution, we report the following activation statistics for condition (1):
>
> |Metric|HotpotQA|Bamboogle|BrowseComp+|MBPP|
> |:--:|:--:|:--:|:--:|:--:|
> |Activation Frequency|1.88|2.70|5.83|3.70|
> |Rectification Success Rate|61%|74%|51%|85%|
>
> We observe that more challenging datasets such as BrowseComp+ and tasks prone to frequent code execution errors such as MBPP naturally lead to higher activation frequencies. Notably, rectification on coding tasks exhibits a particularly strong success rate.
>
> To make these behaviors more concrete, we have already provided examples of rectifier corrections for code execution failures in Appendix D.1 and D.2, and in the revised manuscript we further include rectifier case studies for BrowseComp+ in Appendix D.3 and D.4. These examples are intended to give readers a clearer view of the rectifier's real-time corrective dynamics.
>
> Taken together, the results demonstrate that the rectifier actively participates across diverse tasks and provides meaningful and effective corrections.

---

> ### Author Response · Authors · 2025-11-22
> **[Part 2/2] Author Response**
>
> > **`Question 1`**: While the paper demonstrates MAS2's effectiveness on academic benchmarks, it lacks real-world deployment or interaction scenarios. Is it possible to verify the effectiveness of the method in a real-world dynamic environment, or to provide some examples of real users using multi-agent systems?
>
>
> We would like to respectfully clarify that the **BrowseComp+** benchmark evaluated in Table 1 is in fact a strong representative of *real-world dynamic tasks*. The benchmark is designed to simulate human research queries, where an agent must browse and synthesize information from multiple real-world webpages and documents before arriving at an answer. Critically, the deep-research capability assessed by BrowseComp+ is precisely the type of functionality targeted by mature commercial systems such as **Google Gemini Deep Research**, **OpenAI Deep Research**, and **Perplexity Deep Research**. We therefore sincerely believe that this benchmark meaningfully reflects the deployment conditions.
>
> To further highlight this real-world nature, the manuscript includes (i) the multi-agent architectures MAS² designs for deep-research tasks (Figure 5 and Appendix C), and (ii) concrete case studies in Appendix D showing how MAS² actively corrects real-world tool failures such as context-overflow errors, long-running execution failures, and Pydantic parsing errors.
>
> Taken together, we humbly believe these results demonstrate that MAS² is already effective in the type of dynamic environments the reviewer is concerned about. We are sincerely grateful for this valuable question.

---

> ### Author Response · Authors · 2025-11-25
> **Thank you & Looking forward to further discussion**
>
> Dear Reviewer **Bhu1**,
>
> We truly appreciate your thoughtful evaluation and insightful feedback! To support clearer communication, we summarize your key concerns and our corresponding clarifications below:
>
> 1. **Regarding the practical complexity of MAS² `Weakness 1`**: We clarified the full training and inference costs, showing that MAS² remains lightweight—requiring only modest GPU hours and adding minimal latency relative to baseline systems.
> 2. **Regarding quantifying rectifier effectiveness `Weakness 2`**:  We supplemented rectifier's activation-frequency and success-rate statistics across datasets.
> 3. **Regarding real-world deployment scenarios `Question 1`**: We respectfully clarified that BrowseComp+ is explicitly designed as a real-world dynamic deep-research benchmark, and we provided concrete case studies showing MAS² handling real-world tool failures.
>
> For other issues not mentioned here, please refer to our detailed rebuttal response. We sincerely hope this addresses your concerns, and we warmly welcome further discussion.
>
> Warm regards,
> **Authors**

---

> > ### Comment · Reviewer_UFEM · 2025-11-27
> > **Reviewer Response**
> >
> > Thanks for the detailed response. I think some of my concerns have been addressed, and I will raise my score.

---

> ### Comment · Reviewer_Bhu1 · 2025-11-28
>
> Thank you for your responses and the additional experimental results, which make this work more comprehensive. I have no further questions.

---

> > ### Author Response · Authors · 2025-11-28
> > **Reviewer Response**
> >
> > Thank you for your time and reponse！

---

### Official Review · Reviewer_u18B · 2025-11-01

**Soundness:** 3
**Presentation:** 3
**Contribution:** 3
**Rating:** 4
**Confidence:** 3

**Summary:**

This paper introduces a novel paradigm for multi-agent systems (MAS) that enables recursive self-generation, automatic configuration, and real-time rectification to handle real-world uncertainties beyond traditional "generate-once-and-deploy" approaches.

Contributions include:

- Shifts from the traditional "generate-once-and-deploy" paradigm to a dynamic process that generates and adjusts MAS configurations iteratively, enabling adaptation to real-world uncertainties.
- Introduces a three-meta-agent architecture: Generator for creating high-level workflow templates, Implementer for assigning LLMs to roles, and Rectifier for real-time monitoring and fault correction.
- Proposes the Collaborative Tree Optimization (CTO) framework, an offline reinforcement learning method using decision trees, reward propagation, and preference alignment to train and optimize the meta-agents for enhanced performance and efficiency.

**Strengths:**

- A paradigm for dynamic and scalable MAS: A self-generative, adaptive multi-agent system that evolves iteratively is presented as an alternative to existing MAS frameworks, aiming to address their noted limitations in scalability and adaptability.

- Generalization via CTO: The experimental results report that meta-agents optimized through the CTO framework exhibit generalization to unseen LLMs, with performance improvements of up to 15.1%, suggesting a degree of practical transferability.

- A concise three-agent architecture: The system is built around a three-meta-agent design (Generator, Implementer, and Rectifier), which is characterized by its modularity and is proposed to be amenable to future extensions.

**Weaknesses:**

Writing
- In section 3.1 you defined $\mathcal{T}$ as the set of available tools, but in section 3.2 $\mathcal{T}$ means the trajectory set
- To better assess the reward function design in Section 3.2, a more complete definition of $C(\tau)$ ("raw resource consumption") would be beneficial, as the current article and its appendix lack the necessary details and examples.

Understanding of the paper:

- The lack of information regarding the training datasets undermines the evaluability of the reported benchmark results. I think it would be beneficial if the authors provide a detailed description of the source and composition of all problem datasets used in their training data.
- In Table 2 of Section 4.3, the score of 68% achieved by Gemini 2.5 Pro on the MATH benchmark appears notably low. Given that the official Gemini reaches 21.6 on HLE and 88 on AIME [1], there is a discrepancy that requires clarification. Further details regarding the evaluation of Gemini 2.5 Pro need to be clarified.

[1] https://deepmind.google/models/gemini/pro/

**Questions:**

- Section 3.2 notes that training the CTO requires efficient value function computation. I am concerned that the current approach to preference dataset construction may not be scalable. I suggest providing a more detailed explanation of the sampling strategy to clarify how computational overhead was managed, especially with an eye toward scaling the dataset size.
- In section 3.2 equation 11, the role of \delta V in adjusting the per-sample loss requires further clarification. It would be important to understand the theoretical motivation or mathematical derivation behind this design.

---

> ### Author Response · Authors · 2025-11-22
> **[Part 1/2] Author Response**
>
> We would like to express our deepest respect for your meticulous review! In response to your thoughtful comments, we have carefully prepared a point-by-point reply:
>
> ---
> > **`Weakness 1`**: In section 3.1 you defined $\mathcal{T}$ as the set of available tools, but in section 3.2 $\mathcal{T}$ means the trajectory set
>
> Thank you for your meticulous reading, and we sincerely apologize for any confusion caused. We have carefully reviewed and revised the manuscript: the tool set in Eq. (1) remains defined as $\mathcal{T}$, and in Section 3.2 (Eq. 9) we now use $\mathcal{TS}(v)$ to denote the set of trajectories traversing $v$.
>
>
>
> ---
> > **`Weakness 2`**: To better assess the reward function design in Section 3.2, a more complete definition of $C(\tau)$ ("raw resource consumption") would be beneficial, as the current article and its appendix lack the necessary details and examples.
>
> We sincerely thank the reviewer for highlighting this critical point! In response, we provide a more detailed formulation of $C(\tau)$ below:
>
> For a trajectory $\tau$, let $\mathcal{C}(\tau) = \{(r_i, b_{j(i)}, \kappa_i)\}$ denote the multiset of all backbone invocations during execution, where role $r_i$ invokes backbone $b_{j(i)}$ with token usage $\kappa_i = (\kappa_i^{\text{in}}, \kappa_i^{\text{out}})$. Each backbone $b_j \in \mathbb{L}$ is associated with an external API pricing function $\psi_{b_j}(\kappa) = \alpha_j \kappa^{\text{in}} + \beta_j \kappa^{\text{out}},$ where $\alpha_j, \beta_j$ are the model-specific per-token rates.
>
> We define the resource consumption $C(\tau)$ as the cumulative API expenditure incurred along $\tau$:
>
> $$
> \boxed{
> C(\tau) = \sum_{(r_i, b_{j(i)}, \kappa_i) \in \mathcal{C}(\tau)} \psi_{b_{j(i)}}\left(\kappa_i\right).
> }
> $$
>
> Section 3.2 then normalizes $C(\tau)$ across trajectories to compute the cost-sensitive reward $R(\tau)$.
>
> ---
> > **`Weakness 3`**: The lack of information regarding the training datasets undermines the evaluability of the reported benchmark results.
>
> To address your concern, we provide a precise breakdown of the training and test splits, along with dataset sources:
>
> | Dataset| Train | Test | Source |
> |-|-|-|-|
> | HotpotQA|1000|300 | Solaris99/AgentBank/HotpotQA|
> | Bamboogle|–| 125 | chiayewken/bamboogle|
> | NQ| 1100| 300|sentence-transformers/natural-questions |
> | BrowseComp+|664|166| texttron/BrowseComp-Plus |
> | MBPP|700|273| Muennighoff/mbpp|
> | MATH| 484|121|hendrycks/math |
> | HumanEval|–| 164|openai/openai_humaneval|
>
> As described in Section 3.2, our data-curation pipeline is applied exclusively to the training split of each dataset in order to construct preference trajectories and train MAS². All evaluations are then conducted strictly on the corresponding test split. We hope this clarification further illuminates our experimental setup; these details have been supplemented in Appendix E.2 of our updated manuscript.
>
>
> ---
> > **`Weakness 4`**: In Table 2 of Section 4.3, the score of 68% achieved by Gemini 2.5 Pro on the MATH benchmark appears notably low. Given that the official Gemini reaches 21.6 on HLE and 88 on AIME [1], there is a discrepancy that requires clarification.
>
> We sincerely apologize for the confusion. Upon careful re-inspection, we discovered that the evaluation prompt originally used for the small-/medium-scale models in Table 1 was not fully appropriate for the larger thinking models evaluated in Table 2. After adapting the evaluation prompt, we now obtain performance that matches their documented capabilities (Gemini-2.5-Pro: 92.2%, GPT-5-Mini: 87.5%). We have likewise updated the MAS² results under this corrected evaluation setup, with MAS² achieving up to 95.8%. The full set of revised results is reported in Table 2 of the updated manuscript.
>
> Overall, these corrections further confirm that MAS² can reliably assemble off-the-shelf LLMs (whether small or large) into multi-agent systems whose performance surpasses that of any individual model. We are deeply grateful to the reviewer for bringing this important issue to our attention.

---

> ### Author Response · Authors · 2025-11-22
> **[Part 2/2] Author Response**
>
> > **`Question 1`**: Section 3.2 notes that training the CTO requires efficient value function computation. I suggest providing a more detailed explanation of the sampling strategy to clarify how computational overhead was managed, especially with an eye toward scaling the dataset size.
>
> Thank you for your insightful question. To address your concern, we (1) provide a complete description of the data-curation process in Section 3.2, and (2) clarify why the size of the CTO-constructed dataset is fully controllable.
>
> For each task query, the CTO sampling pipeline expands only a *fixed-width, depth-3* decision tree:
>
> * **Generator:** produces $K$ workflow templates
> * **Implementor:** each template is expanded into $N$ instantiations
> * **Rectifier:** optionally corrects failing instantiations without increasing tree depth
>
> Thus, the total number of sampled trajectories is strictly $|\mathcal{T}| = K\cdot N$. From these trajectories, the preference data for each meta-agent admits tight upper bounds:
>
> * Generator: at most $\binom{K}{2}$ preference pairs
> * Implementor: at most $K\cdot \binom{N}{2}$ pairs
> * Rectifier: at most $K\cdot N$ pairs
>
>
>
> As a result, for a given query, executing $K\cdot N$ trajectories actually yields a total amount of preference data equal to $(K+1)\cdot \binom{N}{2} + K\cdot N$, which exceeds $K\cdot N$ itself.  These observations lead us to believe that constructing data for CTO does not require excessive trajectory rollouts and that the framework scales effectively.
>
> To further support this claim, we have added results showing the wall-clock time for data collection, the resulting data volume, and the performance under different values of $K$ and $N$:
>
>
> |Setting|Total traj. rollouts|#Generator data|#Implementor data|#Rectifier Data|Data collection wall-clock time|HotpotQA (Acc.)|Bamboogle (Acc.)|NQ (Acc.)|
> |-|-|-|-|-|-|-|-|-|
> |$K=2,N=2$|1200|606|1009|1130|8.3h|79.3|57.0|73.7|
> |$K=2,N=4$|2400|609|10235|2267|17.2h|85.6|65.6|75.0|
> |$K=4,N=2$ (original)|2400|1583|2371|2189|14.8h|87.0|64.8|77.0|a
>
>
> As expected, increasing either $K$ or $N$ expands the amount of collected preference data, which in turn yields consistent performance improvements. We also note that MAS² remains computationally reasonable. For example, under the setting $K{=}2, N{=}4$, **2.4K** trajectory rollouts already produce more than **13K** training data. We believe this illustrates the strong scaling potential of MAS² in a practical and efficient manner.
>
>
>
> ---
> > **`Question 2`**: In section 3.2 equation 11, the role of \delta V in adjusting the per-sample loss requires further clarification. It would be important to understand the theoretical motivation or mathematical derivation behind this design.
>
>
> Thank you for raising this insightful question. We would like to address it from two perspectives: **mathematical intuition** and **support from prior literature**.
>
>
> **Mathematical intuition**
>
> The role of $\Delta V$ follows from the value annotation of each decision node $v$ in the curated tree $\mathcal{G}_Q$. Each preference tuple compares two successor nodes $v'$ and $v''$ with a *value gap* $\Delta V = V(v') - V(v'') > 0$. The objective is to update the policy so that actions leading to higher-value successors are preferred in proportion to how much better they are.
>
> A natural surrogate is the value-weighted preference likelihood
> $$
> \mathcal{L}\_{\text{CTO}}(\pi\_\theta)
> = -\mathbb{E}\_{(c\_v,a\_{\text{win}},a\_{\text{lose}},\Delta V)}
> \Big[\Delta V \cdot
> \log \sigma\big(\beta\, r\_\theta(c\_v,a\_{\text{win}})-\beta r\_\theta(c\_v,a\_{\text{lose}})\big)\Big],
> $$
> where $r\_\theta(c\_v,a)$ is the relative preference score. Here, $\Delta V$ acts as a preference-intensity weight: larger value gaps indicate clearer, more consequential distinctions.
>
> The gradient takes the form
> $$
> \nabla\_\theta \mathcal{L}\_{\text{CTO}}
> \propto \mathbb{E}[\Delta V \cdot g\_\theta(c\_v,a\_{\text{win}},a\_{\text{lose}})].
> $$
> Pairs with small $\Delta V$ induce smaller updates, while high-margin pairs receive proportionally stronger emphasis. This encourages the policy to respect both the ordering and the *magnitude* of value differences, improving stability and directing learning toward the most informative decisions.
>
>
>
> **Support from prior literature**
>
> Value-guided preference optimization of this form is also supported by prior work [1,2] (and we directly acknowledge this on Line 269 of our manuscript). For example, Rebel [1] incorporates the *reward difference* between two responses directly into the preference loss. These efforts demonstrate that leveraging graded value gaps, not just binary preferences, yields more informative and stable optimization. Subsequently, CTO adopts this well-established principle.
>
> ---
>
> [1] Rebel: Reinforcement learning via regressing relative rewards
> [2] Relative Preference Optimization: Enhancing LLM Alignment through Contrasting Responses across Identical and Diverse Prompts

---

> ### Author Response · Authors · 2025-11-25
> **Thank you & Looking forward to further discussion**
>
> Dear Reviewer **u18B**,
>
> We sincerely appreciate your detailed reading and constructive feedback. To facilitate clearer communication, we summarize several key concerns and our corresponding clarifications below:
>
>
> 1. **Regardin dataset details `Weakness 3`**: We supplemented explicit training/test splits and dataset sources.
> 2. **Regarding CTO's data scalability `Question 1`**: We detailed the CTO sampling pipeline and provided full scaling experiments showing practical data efficiency across different $K,N$ settings.
> 3. **Regarding the role of (\Delta V) in Eq. (11) `Question 2`**: We clarified its mathematical motivation as a value-gap weighting term, supported both by theoretical intuition and prior literature.
>
> For other issues not mentioned here, please refer to our detailed rebuttal response. We warmly appreciate your effort and look forward to further discussion.
>
> Warm regards,
> **Authors**

---

### Comment · Area_Chair_1Ab3 · 2025-11-27
**Please review the authors' responses and provide feedback ASAP**

Dear Reviewers,

Thank you for your essential contributions to the review process. The authors have submitted their responses to your initial reviews.

I kindly ask you to carefully review the authors' responses for this submission. Your timely assessment of how the authors have addressed your original concerns is a critical step in reaching a final decision.

Please provide your feedback and any necessary updates to your reviews as soon as possible to ensure we can meet our tight schedule for the discussion phase.

Your prompt attention to this matter is highly appreciated.

Regards,

-AC

---

### Meta-Review · Area_Chair_gyCw · 2026-01-01

**Summary:**

This paper introduces a novel multi-agent system (MAS) framework with self-generation, self-configuration, and self-correction to generate multi-agent systems. All reviewers acknowledge the novelty and quality of this paper. Specifically, after the reviewing stage, reviewers still have several concerns as:

1. Lacking experimental details (Reviewer **u18B**, **UFEM**, **3oGm**)
2. Inconsistent Results (Reviewer **u18B**, **UFEM**)
3. Scalability (Reviewer **u18B**)
4. Role of ($\Delta V$), and ($R_p(\tau)$) and ($\theta_C$)  (Reviewer **u18B**, **UFEM**, **3oGm**)
5. Real-world Application (Reviewer **Bhu1**, **3oGm**)
6. Dynamic interference simulations (Reviewer **3oGm**)

For these questions, authors also provided thorough responses and also further experimental results to address reviewers' concerns. Considering the positive feedback from reviewer (Reviewer **UFEM** also promise to increase score), I recommend this paper as a acceptance, while authors also need to merge these reviewers' suggestions into the final version.

**Reviewer Concerns:**

Authors have offer more details about experimental details, addressing inconsistent results, data scalability, more explainations about some variables and ablation studies. Most of reviewers' concerns will be addressed.

**Reviewer Scores:**

Reviewer **UFEM** has promised to increase his score as authors have provided sufficient feedback to address their concerns.

---

### Decision · Program_Chairs · 2026-01-26

Accept (Poster)